# Genome-Wide Identification and Expression Analysis of the *Ginkgo biloba* B-Box Gene Family in Response to Hormone Treatments, Flavonoid Levels, and Water Stress

**DOI:** 10.3390/ijms26178427

**Published:** 2025-08-29

**Authors:** Meiling Ming, Mulin Yi, Kexin Sun, Anning Zu, Juan Zhang, Fangfang Fu, Fuliang Cao, Xiaoming Yang

**Affiliations:** State Key Laboratory of Tree Genetics and Breeding, Co-Innovation Center for Sustainable Forestry in Southern China, Nanjing Forestry University, Nanjing 210037, China; mingmeiling@njfu.edu.cn (M.M.); yimumu@njfu.edu.cn (M.Y.); kexinsun@njfu.edu.cn (K.S.); zuanning@njfu.edu.cn (A.Z.); juanzhang@njfu.edu.cn (J.Z.); fffu@njfu.edu.cn (F.F.)

**Keywords:** *Ginkgo biloba*, BBX, gene family, hormones, flavonoid regulation, water stress

## Abstract

B-box (BBX) transcription factors, which are specific to the plant kingdom, play a crucial role in regulating light-dependent growth, development, secondary metabolite biosynthesis, and the response to biotic and abiotic stresses. Despite their significance, there has been a lack of systematic investigation into the BBX gene family in *Ginkgo biloba*. In the present study, we identified nine BBX genes within the *G. biloba* reference genome, distributed across seven chromosomes, and classified them into four groups based on their phylogenetic relationships with the BBX gene families of *Arabidopsis thaliana*. Our analysis of gene structure, conserved domains, and motifs suggests that *GbBBX*s exhibit a high degree of conservation throughout evolutionary history. Additionally, synteny analysis revealed that dispersed duplication events have contributed to the expansion of the BBX gene family in *G. biloba*. An examination of cis-regulatory elements indicated that numerous *GbBBX* genes contain motifs associated with light, hormones, and stress, suggesting their potential roles in responding to these signals and environmental adaptation. Expression profiles obtained from RNA-Seq data and quantitative Real-Time PCR (qRT-PCR) analyses of *GbBBX* genes across various organs, hormone treatments, and leaves with differing flavonoid content, as well as during both short-term and long-term water stress, demonstrated their potential roles in flavonoid regulation and responses to hormones and water stress. Subcellular localization studies indicated that the proteins GbBBX5, GbBBX7, GbBBX8, and GbBBX9 are localized within the nucleus. This study is the first thorough analysis of the BBX gene family in *G. biloba*, providing a valuable foundation for further understanding their evolutionary context and functional roles in flavonoid regulation and responses to water stress.

## 1. Introduction

Transcription factors (TFs) are a category of proteins characterized by structural domains that facilitate binding to specific DNA sequences found within gene promoter regions. These proteins function as molecular regulators, modulating the transcriptional activity of various genes by either promoting or repressing their expression [1]. Prior research has delineated various families of transcription factors, encompassing R2R3-myeloblastoma (MYB), basic helix-loop-helix (bHLH3), WD40, APETALA2/ethylene-responsive factor (AP2/ERF), basic leucine zipper (bZIP), MADS-box, WRKY, NAC (NAM, ATAF1/2, and CUC2), homeodomain-leucine zipper (HD-Zip), and SQUAMOSA promoter-binding protein-like (SPL) transcription factors [2]. The zinc finger transcription factor family represents one of the most extensive groups of transcription factors found in eukaryotic organisms. This family can be categorized into various subfamilies based on the structural and functional characteristics of its constituents. Among these subfamilies is the B-box (BBX), which plays a crucial role in the regulation of light-dependent plant growth and development [3]. BBX proteins typically possess one or two B-box domains located at the N-terminus, each containing a zinc-binding motif [3]. Additionally, the C-terminus of these proteins is frequently associated with a CCT domain, which stands for CONSTANS, CO-LIKE, and TOC1 domains [3,4]. The B-box domain facilitates interactions among BBX proteins and with a range of other proteins, whereas the CCT domain primarily plays a role in the nuclear localization of BBX proteins. Furthermore, certain BBX proteins possess a valine-proline (VP) motif, which facilitates a distinct interaction with the COP1 (CONSTITUTIVELY PHOTOMORPHOGENIC 1) protein [5].

The first identified BBX gene in plants, CONSTANS (CO), was discovered in *Arabidopsis* in 1995, and plays a crucial role in the regulation of photoperiodic flowering [6]. With the advent of complete genomic sequences for various plant species, a significant number of BBX genes have been isolated across multiple plants. For instance, 32 BBX genes have been identified in *Arabidopsis* [4], 30 in rice [7], 29 in tomato [8], 25 in pear [9], 64 in apple [10], 19 in *Dendrobium officinale*, 16 in *Phalaenopsis equestris* [11], 17, 18, 37 and 33 in *Gossypium arboreum*, *G. raimondii*, *G. hirsutum* and *G. barbadense* [12], 51 in strawberry [13], 25 in grape [14,15], 27 in Moso bamboo [16], 28 in Tartary buckwheat [17], 20 in *Melilotus albus* [18], 26 in cucumber [19], 59 in soybean [20], 27 in *Artemisia annua* [21], 27 in *Salvia miltiorrhiza* [22], 125 in three *Medicago* Species [23]. Notably, the BBX gene family in *Arabidopsis* has been extensively studied regarding its physiological and molecular functions, and these genes can be categorized into five distinct groups (I-V) based on their domain structures [24].

Under abiotic stress, plants induce the expression of relevant genes and activate complex regulatory networks to mitigate the damage caused by environmental adversity. Genes involved in plant abiotic stress response can be broadly categorized into two groups: functional genes, which directly contribute to osmotic adjustment and cellular protection, and regulatory genes, which enhance abiotic stress tolerance by modulating the expression of downstream target genes. For example, overexpression of *MbMYBC1* in *Arabidopsis thaliana* enhances tolerance to cold and drought stress by elevating antioxidant enzyme activities (CAT, POD, SOD), increasing proline content and electrolyte leakage, reducing chlorophyll, and upregulating stress-responsive genes including *AtDREB1A*, *AtCOR15a*, *AtERD10B*, *AtCOR47*, *AtSnRK2.4*, *AtRD29A*, *AtSOD1*, and *AtP5CS1* [25]. Overexpression of *MbICE1* in *Arabidopsis* conferred enhanced tolerance to drought and cold stresses, accompanied by increased chlorophyll and proline content, reduced malondialdehyde (MDA) and electrolyte leakage (EL), attenuated reactive oxygen species (ROS) accumulation, elevated antioxidant enzyme activity, and upregulation of stress-responsive genes including *AtCBF1*, *AtCBF2*, *AtCBF3*, *AtCOR15a*, *AtCOR47*, and *AtKIN1* [26]. Overexpression of either *FvMYB44* or *FvMYB114* in *Arabidopsis thaliana* significantly enhanced tolerance to salt and cold stress, increasing proline and chlorophyll content as well as antioxidant enzyme activities (SOD, POD, CAT), while reducing MDA and ROS levels compared to wild-type and unloaded lines [27,28]. Furthermore, the *EjWRKY17* gene in loquat (*Eriobotrya japonica*) enhances drought tolerance in transgenic *Arabidopsis* by mediating stomatal closure through activation of the ABA signaling pathway [29].

BBX proteins serve as crucial components within regulatory networks that govern light-dependent growth and developmental processes, encompassing seedling photomorphogenesis, the photoperiodic regulation of flowering, hypocotyl elongation, shade avoidance mechanisms, and responses to biotic and abiotic stressors [3]. The BBX family members, including *AtBBX1*, *AtBBX4/COL3*, *AtBBX6/COL5*, *AtBBX7/COL9*, *AtBBX10/COL12*, and *AtBBX17/COL8*, play a crucial role in the regulatory networks that control the processes of floral transition and flower development in *Arabidopsis*. [30]. Additionally, the transcript levels of various BBX genes in crops, including *OsBBX2*, *OsBBX7*, *OsBBX14*, and *IbBBX24*, increase following exogenous application of phytohormones such as abscisic acid (ABA), gibberellin (GA), jasmonic acid (JA), and salicylic acid (SA) [31,32]. Several apple BBX proteins, including MdBBX1, MdBBX22, and MdBBX33/MdCOL11, act as positive regulators of anthocyanin synthesis [33], while MdBBX37 is identified as an inhibitor of this process [34]. For example, low temperatures and UV-B light are linked to the upregulation of *MdBBX33* expression, enhancing anthocyanin accumulation in apple fruits [35]. Similarly, two BBX proteins, PpBBX16 and PpBBX18, have been recognized as positive regulators of anthocyanin accumulation in red pear [36,37]. However, the PpBBX21 protein directly interacts with PpBBX18 or PpHY5, preventing the formation of the PpBBX18-PpHY5 complex and subsequently repressing anthocyanin biosynthesis. Anthocyanins are responsible not only for the red to black color of plant flowers and fruits but are also linked to biotic and abiotic stresses, including viral pathogens, wounding, and drought, while providing protection against photooxidative and heat damage [38]. Numerous reports suggest that BBX proteins play a role in the signaling pathways activated by abiotic stresses such as low temperature, high salinity, drought, and heat. In *Arabidopsis*, *AtBBX18* reduces thermotolerance by regulating genes that respond to heat stress, while *AtBBX24/STO* enhances root growth in high-salt environments [39]. Overexpressing *MdBBX10* in *Arabidopsis* boosts salt and drought tolerance [40], while *AtBBX29* overexpression in *Saccharum* enhances drought resilience [41]. Additionally, *OsBBX8* overexpression in *Oryza sativa* reduced drought tolerance [42], while the overexpression of *GbBBX25* enhances salt tolerance in transgenic poplar [43].

BBX transcription factors are acknowledged as key regulators of various secondary metabolites and stress response pathways in angiosperms, including woody plants [9,10,14,15]; however, their functional characterization in gymnosperms such as *G. biloba* has not been extensively explored. As a paleoendemic species often referred to as a “living fossil,” *G. biloba* is cultivated globally for its medicinal properties and unique fan-shaped leaves. The medicinal benefits of *G. biloba* are primarily attributed to secondary metabolites, including flavonoids and terpenoids. Furthermore, *G. biloba* exhibits remarkable resilience to biotic and abiotic stressors, with notable drought tolerance linked to its evolutionary history. This study identified nine putative members of the BBX gene family within the *G. biloba* genome. A comprehensive bioinformatics analysis was conducted, focusing on the physicochemical properties of the genes, phylogenetic relationships, gene structure, species collinearity, cis-acting elements, and expression patterns of *GbBBX* genes. Subsequently, genes associated with hormone signaling, flavonoid biosynthesis, and water stress were selected for further examination. The findings of this research offer novel perspectives on the possible functions of the *G. biloba* BBX gene family in the regulation of flavonoid metabolism and responses to water stress. This work will be beneficial for future functional investigations and molecular breeding initiatives in *G. biloba* and other forest tree species.

## 2. Results

### 2.1. Whole-Genome Identification of the GbBBX Genes in G. biloba

The BBX gene family in *G. biloba* was systematically analyzed utilizing two distinct methodologies: the BLAST approach and the HMMsearch approach. The BLAST method yielded the identification of 28 candidate genes (Appendix A), while the HMMsearch method identified only 10 candidate genes (Appendix A). Among these, nine genes were found to overlap between the two methods and were designated as *GbBBX* genes in *G. biloba* (Table 1). Following this, the genes with incomplete SBP domains were excluded based on information from the NCBI Conserved Domain Database (CDD). Ultimately, nine *GbBBX* genes possessing the conserved B-Box domain were identified in *G. biloba* and renamed according to their chromosomal locations (Table 1). Notably, *GbBBX5* (also designated as *GbBBX25*) has been demonstrated to be involved in salt tolerance. It was named *GbBBX25* due to its high homology with *Arabidopsis thaliana* AtBBX25 [43]. Notable variability was observed in the properties of the proteins: the lengths of the amino acid sequences ranged from 248 amino acids (GbBBX3) to 536 amino acids (GbBBX9), with corresponding molecular weights varying from 26,808.51 Da to 60,029.39 Da (Table 1). The isoelectric points (pI) of the proteins ranged from 4.84 (GbBBX3) to 8.11 (GbBBX4). Furthermore, the grand average of hydropathicity indices varied from −0.746 to −0.173, and the instability and aliphatic indices were also evaluated. The instability index ranged from 41.33 (GbBBX1) to 56.35 (GbBBX5), indicating that all GbBBX proteins are classified as unstable. Bioinformatic analysis predicted nuclear localization for all BBX proteins except GbBBX4 and GbBBX6, which were targeted to chloroplasts.

The chromosomal mapping of nine *GbBBX* genes in *G. biloba* genome demonstrated a relatively uniform genomic distribution. In total, these nine *GbBBX* genes were located across seven chromosomes: chr1, chr3, chr4, chr5, chr7, chr11, and chr12 (Figure 1). Notably, chromosomes 1 and 5 harbor two *GbBBX* genes, whereas each of the other chromosomes contains a single *GbBBX* gene (Figure 1).

### 2.2. Phylogenetic Analysis of GbBBX Genes

To examine the evolutionary relationships among the identified *GbBBX* genes in *G. biloba*, a Maximum Likelihood (ML) phylogenetic tree was constructed utilizing the protein sequences of nine *GbBBX* genes from *G. biloba*, thirty-two *AtBBX* genes from *Arabidopsis thaliana*, and thirty *OsBBX* genes from rice (*Oryza sativa*) (Figure 2). The phylogenetic tree was classified into five subfamilies (Group I to Group V) according to the established Structure Groups framework defined in *Arabidopsis thaliana* [4], aligning with prior classifications of BBX family distributions [14,22]. The *GbBBX* gene family was found to be distributed across four groups (Groups I, III, IV, and V), comprising two members in Group I, two members in Group III, three members in Group IV, and two members in Group V (Figure 2). These findings imply that the functional roles of BBX genes in group II may have diverged during the evolutionary trajectories of *Arabidopsis thaliana* (a dicotyledon), rice (a monocotyledon), and *G. biloba* (a gymnosperm).

### 2.3. Motif, Conserved Domain, and Gene Structure Analysis of GbBBXs

In order to enhance our understanding of the *GbBBX* gene family, we conducted an analysis of the motifs, conserved domains, and gene structures associated with its members. These genes were clustered based on the results obtained from a phylogenetic analysis of BBX in *G. biloba*, as illustrated in Figure 3A. We employed the MEME web tool to examine the differences among the ten conserved motifs (Motif 1-Motif 10) present in the GbBBX proteins (Figure 3B and Appendix A). It was observed that Motif 1 and Motif 3 were universally present across all nine *GbBBX* genes, while Motif 4 was identified in seven members of the *GbBBX* gene family. Motifs 1 and 3 are likely components of the conserved B-Box1 and B-Box2 domains, and the CCT domain likely consists of Motif 2 (Appendix A). The analysis of conserved motifs revealed that members within the same subgroup of *GbBBX*s tend to exhibit similar motif types, whereas distinct subgroups display variations in motif composition. For instance, both *GbBBX5* and *GbBBX7*, which are classified in group IV, contain motifs 4, 1, and 3; however, their motif arrangements differ significantly from those of group I members, *GbBBX1* and *GbBBX8*. Furthermore, despite being grouped together, *GbBBX6* in group IV demonstrates notable differences in motif composition when compared to its group counterparts, *GbBBX5* and *GbBBX7* (Figure 3B). These discrepancies within the same subgroup imply that genes classified within the same subgroup may possess functional diversity.

The conserved domains of the *GbBBX* gene family members were systematically analyzed. It was observed that the B-Box1 domain is universally present across all nine members of the *GbBBX* family, while the B-Box2 domain is found in only five of these members. Structural conservation was evident within each group, with all members in the same group exhibiting a consistent domain composition. For example, members of Group I, specifically *GbBBX1* and *GbBBX8*, possess the B-Box1, B-Box2, and CCT domains, whereas members of Group V, namely *GbBBX3* and *GbBBX4*, are characterized solely by the presence of the B-Box1 domain (Figure 3C).

Gene structure represents a significant characteristic that can affect gene expression and functionality. In the present study, we analyzed the exon-intron compositions of the nine *GbBBX* genes (Figure 3D). The number of exons (coding sequences, CDSs) varied from 2 in *GbBBX1*, *GbBBX3*, *GbBBX4*, *GbBBX8*, and *GbBBX9* to 7 in *GbBBX6*, while the intron count ranged from 1 to 6. This variability in gene structure suggests a functional diversity among the *GbBBX* genes. Although the core coding sequences are clearly delineated, there exists a notable deficiency in the comprehensive annotation of regulatory elements, such as the 5′-untranslated region (5′-UTR) and 3′-untranslated region (3′-UTR), within the current genomic resources for *G. biloba* [44].

### 2.4. Multiple Sequence Alignment Analysis

Comparative mapping of domain logos and sequences revealed three key elements among the 9 *GbBBX*s: a B-box1 domain, a B-box2 domain, and a CCT domain (Figure 4). The analysis of sequence conservation indicated that the B-box1 and B-box2 domains exhibit strong conservation at the C-X2-C-X8-C-X7-C-X2-C-D-X3-H-X8-H-X4 sequences, comprising about 44 amino acid residues (Figure 4A,B). Furthermore, the consensus sequence for the conserved CCT domain was K-X2-R-Y-X2-R-K-X2-R-K-X2-A-X2-R-X-R-X-K-G-R-F (Figure 4C).

### 2.5. The Expansion and Collinearity Analysis of the GbBBX Gene Family

The MCScanX software (V1.0.0) was employed to investigate the expansion and evolutionary history of the *GbBBX* gene family. Analysis revealed that, among the nine members of the *GbBBX* gene family, seven were derived from dispersed duplication, while two originated from WGD or segmental duplication (Figure 5A). Notably, none of the *GbBBX* genes were produced through singleton, tandem, or proximal duplication (Figure 5A). Additionally, one syntenic gene pair, *GbBBX1*-*GbBBX8*, was identified within the *GbBBX* gene family (Figure 5B). These findings suggest that dispersed duplication events were the predominant mechanism driving the expansion of the *GbBBX* gene family.

To investigate the evolutionary dynamics associated with the duplication of BBX genes, cross-species synteny analyses were performed, comparing *G. biloba* with both a herbaceous model organism, *Arabidopsis thaliana*, and a woody relative, *Populus alba* × *Populus tremula* var. *glandulosa* clone ‘84K’. The analyses employed MCScanX for the detection of whole-genome collinearity. The findings indicate that *G. biloba* exhibits a lower degree of genome-wide covariance with *Arabidopsis thaliana*, and no covariant BBX genes have been identified between these two species. Furthermore, while *G. biloba* does not share covariant BBX genes with the woody plant Poplar, it possesses more homologous gene pairs than *Arabidopsis* across the entire genome (Appendix A). Despite their relatively recent divergence as woody species, *G. biloba* and Poplar display varied architectures of BBX genes, implying a relatively low degree of conservation within this regulatory gene family over time. Comparative analyses of *GbBBX* homologs across species may elucidate patterns of functional divergence and contribute to the reconstruction of ancestral gene repertoires in angiosperms (such as *Arabidopsis thaliana* and *Populus*) and gymnosperms (including *G. biloba*).

### 2.6. Cis-Acting Elements Analysis in the Promoter of GbBBX Genes

Cis-regulatory elements located within promoter regions are instrumental in mediating transcriptional modulation, thereby influencing phenotypes through the precise spatiotemporal regulation of gene activity. An in silico analysis of the promoters of *GbBBX* genes was performed utilizing the PlantCARE database, which involved a systematic examination of cis-acting elements located within 2000 base pairs upstream of transcription start sites. A varied distribution of cis-elements was identified across each *GbBBX* promoter, encompassing categories such as light, auxin, gibberellin, methyl jasmonate (MeJA), abscisic acid, defense and stress responses, as well as low-temperature elements, among others (Figure 6A). This diversity suggests that *GbBBX* genes may participate in a multitude of biological processes. The identified cis-elements were classified into four distinct groups: hormone-responsive elements (including TCA-elements, ABRE, CGTCA motif, TGACG motif, P-box, TGA-element, TATC-box, and AuxRR-core), light-responsive elements (such as I-box, ACE, GATA motif, GT1 motif, chs−Unit 1 m1, G-box, MRE, ATC motif, chs−CMA1a, TCT-motif, TCCC motif, Sp1, LAMP element, AE-box, and Gap box), plant growth and development-responsive elements (including CAT-box, O2-site, and HD−Zip), and stress-responsive elements (such as MBS, ARE, TC-rich repeats, and LTR) (Figure 6B). The predominant category of cis-elements identified was light-responsive, although hormone-responsive and stress-responsive elements were also significantly represented in the promoters of *GbBBX* genes. Plant hormones, including abscisic acid and gibberellin, are acknowledged as critical components in various stress signaling pathways, and the presence of these cis-acting elements in the promoters of *GbBBX* genes contributes to their functional diversity under stress conditions. *GbBBX9* contained the highest density of cis-regulatory elements associated with hormone, light, and stress responsiveness, whereas *GbBBX2* featured the most abundant elements linked to growth regulation. The analysis of cis-acting elements indicates that *GbBBX* genes can be regulated by both light and stress responsiveness.

### 2.7. Expression Profiles of GbBBX Genes in Various Organs, Hormones, Low- and High-Flavonoid Leaves, and Water Stress in G. biloba

Initially, we examined the organ-specific expression profiles utilizing RNA-Seq data derived from the referenced study [45]. The RNA-Seq expression profiles indicated that the *GbBBX* genes exhibited differential expression across eight distinct organs, namely root (R), stem (S), immature leaf (IL), mature leaf (ML), immature fruit (IF), mature fruit (MF), microstrobilus (M), and ovulate strobilus (OS) (Figure 7A). Notably, all nine *GbBBX* genes demonstrated elevated expression levels in the stem, whereas only two *GbBBX* genes, specifically *GbBBX5* and *GbBBX7*, showed significant expression in the root (Figure 7A). The expression patterns of two BBX genes (*GbBBX5* and *GbBBX6*) were markedly higher in mature leaves compared to immature leaves, suggesting their potential regulation in leaf development. Furthermore, the expression patterns of the majority of *GbBBX* genes in both mature and immature fruits were largely consistent, indicating that the expression of these BBX genes is organ-specific rather than specific to the maturation stage of the fruit. Additionally, most *GbBBX* genes, with the exception of *GbBBX5* and *GbBBX6*, exhibited high expression levels in both the ovulate strobilus and the microstrobilus, implying that they may not be regulated in the transition to the reproductive phase.

Then, the transcript abundances of *GbBBX* genes associated with plant hormone responsiveness were examined. To evaluate *GbBBX* expression responses to phytohormones, we analyzed transcriptomic data from Ginkgo leaves treated with abscisic acid (ABA), brassinosteroids (BR), or salicylic acid (SA) in a published dataset [46,47]. Results indicated that five out of nine *GbBBX*s (*GbBBX1*, *GbBBX2*, *GbBBX4*, *GbBBX8*, and *GbBBX9*) exhibited induction at concentrations of 0.5 or 1 mmol/L ABA (ABA_0.5 and ABA_1). However, this induction was diminished when the concentration was increased to 1.5–2 mmol/L (ABA_1.5 and ABA_2) (Figure 7B). Additionally, five *GbBBX*s (*GbBBX1*, *GbBBX2*, *GbBBX5*, *GbBBX8*, and *GbBBX9*) demonstrated upregulated expression in response to BR treatment, particularly at concentrations of 1.5 mg/L (BR_1.5) (Figure 7C). Under a concentration of 1 mmol/L SA (SA_1), several *GbBBX*s, including *GbBBX3*, *GbBBX7*, and *GbBBX8*, were induced to their highest levels; however, as the concentration increased to 3 mmol/L (SA_3), their expression was downregulated in comparison to SA_1 (Figure 7D). Notably, *GbBBX8* was induced by all three exogenous treatments of ABA, BR, and SA, suggesting it can be up-regulated by cross-phytohormone responsiveness.

Next, we explored the expression profiles of GbBBXs in four groups contrasting low and high flavonoid content [48,49,50,51]. Group I consists of Ginkgo leaves exposed to UV-B-free white light, characterized by low flavonoid content (Leaf_WL), in contrast to Ginkgo leaves subjected to prolonged UV-B exposure, which exhibit high flavonoid content (Leaf_UV). Group II includes leaves with relatively low flavonoid content (GB_FL) compared to those with relatively high flavonoid content (GB_FH) sourced from 112 distinct Ginkgo clones. Group III encompasses green Ginkgo leaves with low flavonoid content (Leaf_GL) in comparison to mutant yellow Ginkgo leaves that possess high flavonoid content (Leaf_YL). Finally, Group IV comprises leaves from a high-flavonoid mutant of Ginkgo (Leaf_ZY1) alongside control leaves from Anlu1 (Leaf_Anlu1). The differentially expressed *GbBBX* genes exhibited variability across the distinct groups analyzed. Specifically, *GbBBX2*, *GbBBX8*, and *GbBBX9* were found to be down-regulated in the high-flavonoid leaves within both the first low- and high-flavonoid group (comprising control and UV-B treatment) and the third low- and high-flavonoid group (comprising green and yellow-mutant leaves), while no significant changes or up-regulation were observed in the other two groups (Figure 8A). Furthermore, *GbBBX7* demonstrated up-regulation in the high-flavonoid leaves under UV-B treatment in the low- and high-flavonoid group I, yet it was down-regulated in the high-flavonoid leaves of the yellow mutant in the group III. In contrast, *GbBBX3* and *GbBBX4* were uniquely up-regulated in the high-flavonoid leaves of the low- and high-flavonoid group IV (‘Anlu1’ and ‘ZY1’ leaves).

Finally, we analyzed the expression patterns of *GbBBX* genes in response to both short-term and long-term water stress. Under short-term water shock and subsequent rehydration (measured at 0, 3, 6, 12, 24 h, and 12 h post-rehydration) [52], the *GbBBX* genes were categorized into two distinct clusters: one cluster exhibited induction in response to water stress (comprising *GbBBX5*, *GbBBX7*, *GbBBX8*, and *GbBBX9*), while the other cluster demonstrated repression (including *GbBBX1* and *GbBBX2*) (Figure 8B). In contrast, during the long-term drought treatment (assessed at 0, 15, 22 days, and 3 days post-rewatering) [53], the majority of *GbBBX* genes were found to be repressed by drought stress (specifically *GbBBX1*, *GbBBX2*, *GbBBX6*, *GbBBX7*, *GbBBX8*, and *GbBBX9*), with only *GbBBX5* showing a slight induction under drought conditions (Figure 8C). Notable differences were observed between the gene expression profiles under short-term and long-term water stress, suggesting the presence of distinct regulatory networks that mediate the responses of *G. biloba* to these varying durations of water stress. For example, while *GbBBX7*, *GbBBX8*, and *GbBBX9* were upregulated during short-term water stress, they were downregulated in the context of long-term drought stress; conversely, *GbBBX5* was consistently upregulated in both scenarios (Figure 8B,C).

### 2.8. Quantitative Real-Time PCR (qRT-PCR) Analysis and Subcellular Localization Assay of the GbBBX Genes

To corroborate the findings obtained from previously reported RNA-Seq data, all nine *GbBBX* genes were subjected to quantitative reverse transcription polymerase chain reaction (qRT-PCR) analysis. The results from qRT-PCR were in alignment with those derived from RNA-Seq data. Under time-course dehydration, *GbBBX2* and *GbBBX6* expression decreased steadily over time. In contrast, *GbBBX3*, *GbBBX4*, *GbBBX7*, and *GbBBX8* exhibited transient upregulation at the 3 h timepoint, followed by a decline. Notably, despite subsequent reduction, *GbBBX7* and *GbBBX8* maintained expression levels significantly above 0 h (before-treatment), indicating sustained net upregulation in short-term water shock (Figure 9A). Under long-term drought stress, *GbBBX6* and *GbBBX8* exhibited significantly reduced expression levels relative to controls with regular irrigation at both 15-day and 22-day timepoints, whereas *GbBBX2* only decreased at 22-day timepoints, further supporting the RNA-Seq data (Figure 9B).

To elucidate the subcellular localization of GbBBX proteins, we conducted an analysis of GbBBX5, GbBBX7, GbBBX8, and GbBBX9 through transient expression assays in the leaves of transgenic *Nicotiana benthamiana* containing a nuclear localization protein with mCherry signals. Microscopic examination demonstrated that the fluorescence signal from the control vector, 35S-GFP, was present in both the nucleus and the cytoplasm (displayed at the periphery of the cells due to the huge vacuole). In contrast, the green fluorescent protein (GFP) signals for GbBBX5, GbBBX7, GbBBX8, and GbBBX9 exhibited significant overlap with the nuclear marker NLS-mCherry, thereby indicating their localization within the nucleus (Figure 10). This observation aligns with the predictions generated by the WoLF PSORT online tool (Table 1). Collectively, these results suggest that GbBBX5, GbBBX7, GbBBX8, and GbBBX9 may function as nuclear proteins, potentially acting as transcription factors.

## 3. Discussion

### 3.1. Characteristics and Evolution of BBX Genes in G. biloba

BBX genes serve as essential transcriptional regulators specific to plants, coordinating complex developmental programs that are dependent on light. These programs include the biosynthesis of secondary metabolites, the morphogenesis of leaves and flowers, and the responses to both biotic and abiotic stresses, all of which are mediated by evolutionarily conserved molecular mechanisms [24]. The characterization of BBX genes began with the identification of CONSTANS (CO) in *Arabidopsis* [6], with subsequent phylogenetic analyses revealing conserved orthologs across monocotyledonous [7] (e.g., rice) and dicotyledonous [8,10,15] (tomato, apple, pear, grape, and so on). The functional conservation and divergence of BBX homologs in gymnosperms, which are evolutionarily significant non-flowering seed plants, have not been thoroughly investigated in comparison to angiosperm systems. *G. biloba*, a notable gymnosperm, has not had its BBX gene systematically investigated. In this study, nine *GbBBX* genes from *G. biloba* were identified utilizing a genome-level database derived from the Pfam database (PF00643) and the BLAST protein database (Table 1). The number of *GbBBX* gene family members is lower than that found in *Arabidopsis thaliana* (32 members) [4], tomato (29 members) [8], rice (30 members) [7], and apple (64 members) [10], yet it is comparable to that of sugar beet (*Beta vulgaris*) (10 members) [54]. Furthermore, chromosomal mapping of the nine *GbBBX* genes in *G. biloba* indicated a relatively even distribution across seven chromosomes (Figure 1), which exhibits convergence with the distribution profile documented in cucumber (*Cucumis sativus*) [19]. Phylogenetic analysis revealed that the *GbBBX* gene family is organized into four distinct groups, in contrast to the five groups identified within the AtBBX gene family (Figure 2). This suggests that the functional roles of BBX genes may have diverged during the evolutionary trajectories of *Arabidopsis thaliana* (a dicotyledon) and *G. biloba* (a gymnosperm).

The phylogenomic analysis, which integrates sequence homology, motif examination, gene structure, and molecular evolution patterns, indicates a conserved taxonomic clustering within *G. biloba*. The *GbBBX* genes, categorized within specific phylogenetic clades, exhibit conserved structural frameworks (Figure 3), suggesting that evolutionary selection pressures have influenced the relationship between gene structure variability and functional diversification. Notably, *GbBBX1* and *GbBBX8* display similar motifs and share three common domains (Figure 3B), which may be attributed to whole genome duplication (WGD) or segmental duplication events. Each *GbBBX* gene is characterized by the presence of a B-box1 domain; however, only *GbBBX1*, *GbBBX5*, *GbBBX6*, *GbBBX7*, and *GbBBX8* also contain a B-box2 domain (Figure 3C). Furthermore, only *GbBBX1*, *GbBBX2*, *GbBBX8*, and *GbBBX9* are found to possess the CCT domain, while none of these genes include the VP domain (Figure 3C). A 14-base pair deletion in *PyBBX24*, referred to as PyBBX24^DN14^, enhances anthocyanin biosynthesis in pear and induces a dwarf phenotype in Arabidopsis, tobacco, and tomato plants, in contrast to the standard *PyBBX24* variant. PyBBX24^DN14^ is capable of activating the expression of *PyGA2ox8* by directly binding to its promoter, which leads to the deactivation of bioactive gibberellins (GAs) and a subsequent reduction in plant height. However, the presence of the VP domain in the C-terminus of *PyBBX24* counteracts these effects [55]. Given the conserved role of VP domains in GA signaling and the universal absence of this domain in *GbBBX*s, we hypothesize that GbBBX proteins might regulate *GbGA2ox* activity, a mechanism potentially linked to bioactive GA deactivation and reduced plant height.

We conducted an analysis of the duplication events associated with the *GbBBX* gene within the complete genome of *G. biloba*. Our findings indicate that dispersed duplication is the predominant form of duplication, followed by whole genome duplication (WGD) or segmental duplication; however, we did not observe any singleton, proximal, or tandem duplication events (Figure 5A). In instances of dispersed duplication, the newly formed genes generally do not retain the exact sequences of their progenitor genes, thus primarily demonstrating non-conservative replication characteristics. This observation implies that transposable element-mediated dispersed replication significantly contributes to the expansion of the *GbBBX* gene family. Furthermore, WGD and segmental duplications are known to facilitate gene family amplification in plants [56], and they also play a notable role in the expansion of the *GbBBX* gene family. Additionally, collinearity analysis with *Arabidopsis thaliana* and Poplar (*Populus alba* × *Populus tremula* var. *glandulosa* clone ‘84K’) revealed the absence of homologous genes for *GbBBX* in both *Arabidopsis thaliana* and Poplar (Appendix A).

### 3.2. Expression Patterns and Potential Functions of the GbBBX Genes

Gene expression profiles are typically strongly correlated with biological functions. The cis-regulatory elements located in the promoter regions of BBX genes are intricately associated with both expression profiles and functional roles, as they are primarily involved in the regulation of gene transcription. Research indicates that numerous cis-elements within the promoter regions of BBX genes are linked to various aspects of plant growth and development, as well as responses to light, hormones, and stress (Figure 6). This pattern is consistent with findings in other plant species, such as *Salvia miltiorrhiza* [22] and *Artemisia annua* [21]. To elucidate the potential functions of *GbBBX* genes in plant growth and development, RNA-Seq-based [45] expression patterns across different organs were initially analyzed (Figure 7A). The majority of *GbBBX* genes demonstrated elevated expression levels in the stem, leaves, and fruits, while exhibiting lower expression in the root, thereby suggesting their involvement in light-dependent processes. Plant hormones are signaling molecules that control essential elements of growth, development, and reactions to environmental stress [57]. Some *GbBBX* genes, such as *GbBBX8*, were induced by all three exogenous treatments of ABA, BR, and SA. However, other *GbBBX* genes, including *GbBBX1*, *GbBBX2*, *GbBBX4*, and *GbBBX9*, were induced by two exogenous hormone treatments (Figure 7B–D). Furthermore, some *GbBBX* genes, like *GbBBX6*, were induced by only one exogenous hormone treatment, suggesting a different role of *GbBBX* genes in various plant hormone responsiveness.

Flavonoids represent the principal active constituents of *G. biloba*, and their biosynthetic pathways are modulated by light and hormonal signals [58]. Given that numerous cis-regulatory elements within the promoter regions of BBX genes are linked to light and hormonal responses, we hypothesized that the *GbBBX* genes may potentially participate in the regulation of flavonoid biosynthesis. Notably, *IbBBX29* exhibited high expression levels in the leaves of sweet potato and was significantly upregulated by auxin (IAA). The overexpression of *IbBBX29* resulted in an increase in leaf biomass ranging from 21.37% to 70.94%, an elevation in IAA levels between 12.08% and 21.85%, and a rise in flavonoid accumulation of 31.33% to 63.03% in sweet potato [59]. To elucidate the potential functions of the *GbBBX* genes in flavonoid regulation, we examined expression patterns across groups with varying flavonoid content (Figure 8A). Transcriptional profiling identified several *GbBBX* genes that were differentially expressed among the four groups characterized by low and high flavonoid content (e.g., *GbBBX2*, *GbBBX3*, *GbBBX4*, *GbBBX7*, *GbBBX8*, and *GbBBX9*). However, the limited overlap of differentially expressed *GbBBX* genes across these groups suggests the presence of stimulus-specific regulatory mechanisms. Group I (Light Quality Effect): Leaf_WL vs. Leaf_UV isolates UV-B photoreceptor signaling (GbBBX2, GbBBX8, and GbBBX9). Group II (Genetic Variation Effect): GB_FL vs. GB_FH across 112 ginkgo clones captures constitutive genetic polymorphisms in flavonoid regulators. Group III (Developmental Mutation Effect): Leaf_GL vs. Leaf_YL compares wild-type and chlorophyll-deficient mutants, where DEGs (GbBBX2, GbBBX8, and GbBBX9) correlate with retrograde signaling from plastid-to-nucleus—a mechanism absent in other groups. Group IV (Artificial Selection Effect): Leaf_ZY1 (high-flavonoid mutant) vs. Leaf_Anlu1 highlights breeding-selected transcriptional rewiring (GbBBX3 and GbBBX4). This compartmentalization confirms that GbBBX genes operate within stimulus-specific regulatory modules: photoreception (Group I), genetic background (Group II), organelle-nucleus crosstalk (Group III), and artificial selection (Group IV).

The regulatory roles of BBX genes in light-dependent plant growth and development have been extensively characterized [30]. However, the involvement of BBX genes in the response to abiotic stress has not been thoroughly investigated. In light of our focus on water stress, we analyzed the expression profiles of all *GbBBX* gene members under both short-term water shock and long-term drought stress conditions (Figure 8B,C). Our findings indicated a decrease in the expression of certain *GbBBX* genes, such as *GbBBX1* and *GbBBX2*, under both short-term and long-term stress conditions. Conversely, *GbBBX5* exhibited consistent upregulation in both scenarios. Notably, the expression patterns of other *GbBBX* genes, specifically *GbBBX7*, *GbBBX8*, and *GbBBX9*, displayed contrasting responses between short-term and long-term drought conditions. This suggests that *GbBBX7*, *GbBBX8*, and *GbBBX9* may facilitate a rapid response to mitigate damage from adverse conditions in the short term, yet they appear to be ineffective during extended periods of drought. The differential responses of genes to short-term versus long-term water stress are consistent with observations in other plant species [60].

## 4. Materials and Methods

### 4.1. Whole-Genome Identification of B-Box Genes in G. biloba

The high-quality genomic data for *G. biloba*, as cited in source [44], were obtained from the Genome Sequence Archive database (https://ngdc.cncb.ac.cn/gwh/Assembly/18742/show, accessed on 16 April 2025). Initially, the amino acid sequences corresponding to the BBX gene family members in *Arabidopsis thaliana* were sourced from the Arabidopsis Information Resource (TAIR) database (https://v2.arabidopsis.org/, accessed on 16 April 2025); the gene IDs are listed in Appendix A. The complete protein sequences of *G. biloba* were employed to create a BLAST protein database utilizing the makeblastdb software (V 2.10.1). Candidate B-Box members were identified under stringent criteria by conducting a BLASTp (V 2.10.1) search against the genome-wide protein database of *G. biloba*, using the Arabidopsis BBX genes as queries (E-value < 1 × 10^−5^, Identity > 30%). Subsequently, Hidden Markov Models (HMMs) associated with the B-Box domain (PF00643) were retrieved from the Pfam database (http://pfam.xfam.org/, accessed on 16 April 2025). The HMMsearch software (V 3.3.2) was then utilized to analyze the complete protein sequences of *G. biloba* in accordance with the Pfam-A models file containing the B-Box domain, which resulted in a list of candidate BBX members for *G. biloba*. Thirdly, overlapping candidate BBX genes were identified by intersecting the results obtained from both the BLAST and HMM searches. Finally, the NCBI CDD website tool (https://www.ncbi.nlm.nih.gov/Structure/bwrpsb/bwrpsb.cgi, accessed on 16 April 2025) was employed to confirm the integrity of the B-Box domains within the *GbBBX* genes.

### 4.2. Physicochemical Properties, Subcellular Localization Prediction, and Chromosomal Location Analysis of BBX Genes

The protein-coding sequence length, molecular weight (MW), and isoelectric point (pI) were determined utilizing the ExPASy ProtParam online tools (https://web.expasy.org/protparam/, accessed on 16 April 2025). Additionally, predictions regarding the subcellular localization of GbBBX proteins were performed using WoLF PSORT (https://wolfpsort.hgc.jp/, accessed on 16 April 2025). Furthermore, the chromosomal localization of the GbBBXs was derived from the genomic annotation data of *G. biloba* and subsequently visualized using TBtools-II (Version 2.210) [61].

### 4.3. Phylogenetic Analysis

Muscle software (V 3.8.1551) was employed to conduct multiple sequence alignments of BBX genes in *Arabidopsis thaliana*, *Oryza sativa*, and *G. biloba*. The gene IDs of rice BBX are also listed in Appendix A. The phylogenetic tree was constructed utilizing IQ-Tree (V 2.1.2), applying the Maximum Likelihood (ML) method alongside 1000 bootstrap iterations. The phylogenetic tree was midpoint-rooted and classified into five subfamilies according to the established Structure Groups framework defined in *Arabidopsis thaliana* [4]. The resultant tree topology was subsequently visualized and annotated using the iTOL web server (https://itol.embl.de/, accessed on 20 April 2025).

### 4.4. Motif, Conserved Domain, Gene Structure, and Cis-Elements Analysis

The MEME Suite (https://meme-suite.org/meme/tools/meme, accessed on 20 April 2025) was employed to identify ten motifs (ranging from 6 to 50 amino acids in width, with any number of repetitions (anr), classic mode) within the BBX family proteins of *G. biloba*. To ascertain the conserved domains present in these BBX family proteins, the NCBI Batch CD-Search Tool (https://www.ncbi.nlm.nih.gov/Structure/bwrpsb/bwrpsb.cgi, accessed on 20 April 2025) was utilized. The gene structure of the GbBBX family was extracted from the genome annotation file of *G. biloba* using TBtools-II (V 2.210). Additionally, the PlantCARE web tool with default settings (https://bioinformatics.psb.ugent.be/webtools/plantcare/html/, accessed on 20 April 2025) was used to predict cis-acting elements located within 2000 base pairs upstream of transcription start sites of the GbBBX genes. Visualization of the data was conducted using TBtools-II (V 2.210).

### 4.5. Gene Duplication and Syntenic Analysis of the GbBBX Gene Family

Genome assemblies and annotation files for *Arabidopsis thaliana* and Poplar (*Populus alba* × *Populus tremula* var. *glandulosa* clone ‘84K’) genome data [62] were retrieved from Ensembl Plants (http://plants.ensembl.org/, accessed on 20 April 2025) and the National Center for Biotechnology Information (NCBI, https://www.ncbi.nlm.nih.gov/, accessed on 1 April 2025). Segmental duplication events between *G. biloba* and *Arabidopsis thaliana* or *Populus* were identified and visualized through synteny analysis utilizing TBtools-II (V 2.210). The replication events of the *GbBBX* genes were examined using multiple collinear scanning toolkits (MCScanX, V 1.0.0). Additionally, the syntenic relationships between the *GbBBX* genes and the BBX genes from *Arabidopsis thaliana* or *Populus* were assessed using the Dual Synteny Plotter software (V 2.210) within TBtools-II (V 2.210).

### 4.6. RNA-Seq and Quantitative Real-Time PCR (qRT-PCR) Based Expression Analysis of GbBBX Genes

The transcriptome data were downloaded from the National Center for Biotechnology Information database (NCBI, https://www.ncbi.nlm.nih.gov/) and then analyzed using the same method as in our previous study [53]. The accession numbers for the various transcriptome datasets utilized in this research are detailed in Appendix A. Heatmaps illustrating the expression analysis of *GbBBX*s were produced using TPM values processed with TBtools-II (version 2.210). Normalized expression data are provided in Appendix A. The heatmap was generated by the averages of three replicates. In accordance with our prior investigations [52,53], a quantitative Real-Time PCR (qRT-PCR) assay was conducted to validate the findings from the transcriptome sequencing. The relative expression levels of transcripts were assessed using the 2^−∆∆Ct^ method, with *GbGAPDH* (evm.model.chr7.1737) serving as the endogenous control gene, which was commonly used in the reported paper on *G. biloba* [58,63,64]. The primer sequences employed in this experiment are listed in Appendix A.

### 4.7. Subcellular Localization Analysis

The full-length coding sequences of *GbBBX5*, *GbBBX7*, *GbBBX8*, and *GbBBX9* (excluding stop codons) were amplified via polymerase chain reaction (PCR) from the cDNA derived from *G. biloba* leaf tissue. The resulting amplified products were subsequently directionally cloned into the pSAK277-GFP binary vector, creating C-terminal GFP fusions regulated by the CaMV 35S promoter. These constructs were then introduced into competent *Agrobacterium tumefaciens* GV3101 cells through a heat-shock transformation method. The analysis of subcellular localization was conducted in accordance with previously established protocols [52]. Transgenic *Nicotiana benthamiana* was engineered to express a fluorescent nuclear marker consisting of mCherry fused to a nuclear localization signal (NLS-mCherry) [65].The abaxial surfaces of transgenic *Nicotiana benthamiana* leaves, aged four weeks, were subjected to agroinfiltration using a bacterial suspension with an optical density at 600 nm (OD600) of 0.6. Subsequently, the leaves were maintained in an incubator at a temperature of 25 °C for a duration of 72 h, after which live-cell imaging was conducted utilizing an inverted confocal microscope (Zeiss LSM 780, ZEISS Microscopy, Oberkochen, Germany). Excitation/emission wavelengths were 480–520 nm for GFP, and 580–620 nm for mCherry. The empty vector 35S-GFP (expressing the GFP reporter protein alone) was injected into the tobacco leaves as controls. The primer sequences utilized in this study are detailed in Appendix A.

### 4.8. Statistical Analysis

Statistical analysis was performed using one-way ANOVA with Tukey’s post hoc multiple comparisons test (*p* < 0.05). Data processing utilized SPSS Statistics (IBM) (V26) and GraphPad Prism v9.0, respectively.

## 5. Conclusions

The *GbBBX* genes underwent a comprehensive investigation of their physicochemical properties, chromosomal locations, phylogenetic relationships, conserved motifs, gene structures, duplication events, evolutionary connections, cis-regulatory elements, and expression profiles. This analysis uncovered that nine *GbBBX*s are located on seven chromosomes and classified into four categories. Synteny analysis revealed that dispersed duplication events have expanded the BBX gene family in *G. biloba*. Expression patterns and molecular functions were examined using previously reported RNA-Seq data and qRT-PCR, demonstrating that different *GbBBX* genes responded variably to hormones, flavonoid regulation, and water stress, either increasing or decreasing. Subcellular localization studies indicated that GbBBX5, GbBBX7, GbBBX8, and GbBBX9 are located in the nucleus. Our results suggest that *GbBBX* genes may play a vital role in flavonoid regulation and responses to hormones and water stress. Overall, our genome-wide analysis of the BBX gene family provides possible molecular targets for precision breeding of *G. biloba*, facilitating the development of drought-resilient cultivars and high-flavonoid varieties for enhanced environmental adaptation and nutraceutical applications.

## Figures and Tables

**Figure 1 ijms-26-08427-f001:**
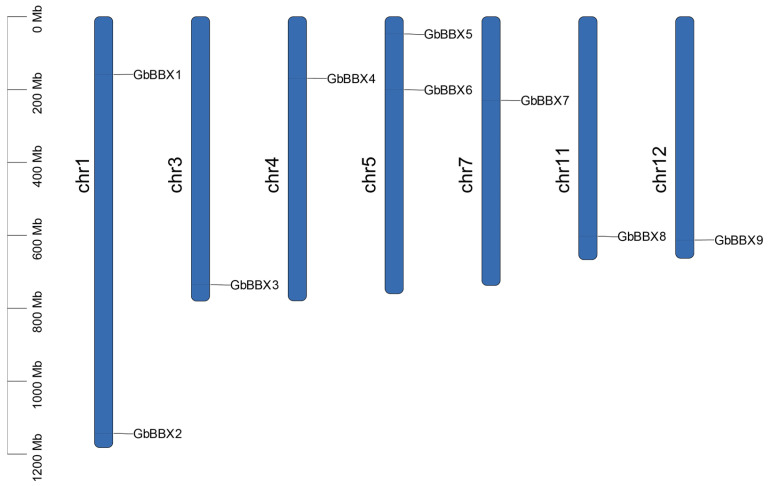
Chromosome distribution of *GbBBX* genes in the genome of *G. biloba*. The length of each chromosome was estimated in megabases (Mb).

**Figure 2 ijms-26-08427-f002:**
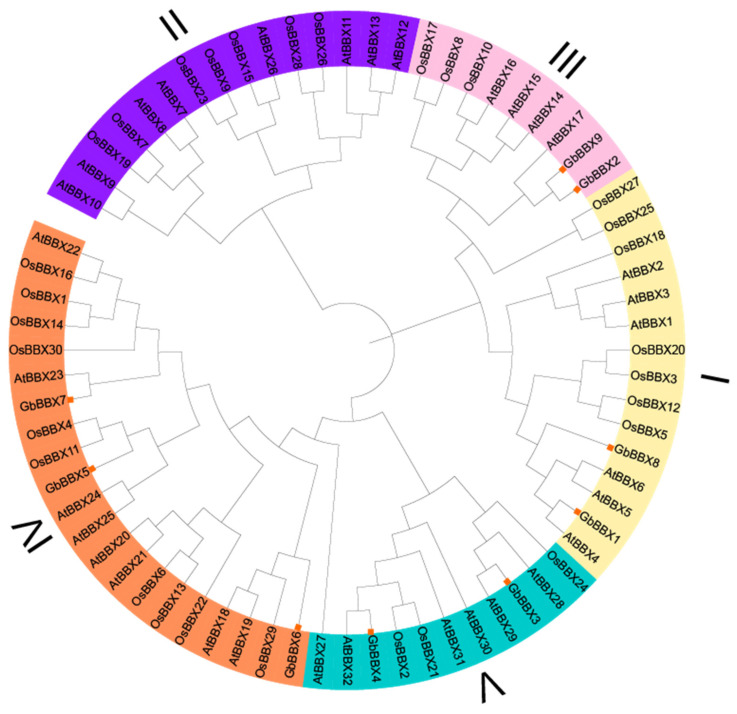
Phylogenetic analysis of GbBBX proteins with AtBBX proteins from *Arabidopsis* and OsBBX proteins from rice (*Oryza sativa*). The phylogenetic tree was constructed using IQ-Tree software (V 2.1.2), employing the Maximum Likelihood (ML) method, and bootstrap replications were set at 1000 times.

**Figure 3 ijms-26-08427-f003:**
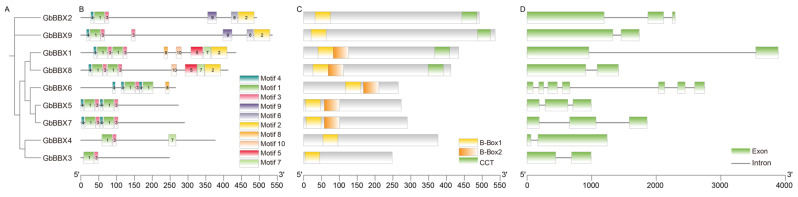
Phylogenetic trees, motifs, conserved domains, and gene structure analysis of GbBBX proteins. (**A**) Phylogenetic trees of GbBBX proteins. (**B**) Motifs in the 9 GbBBX proteins are represented by ten colored boxes, indicating different motifs. (**C**) Conserved domains present in the 9 GbBBX proteins. (**D**) Exon-intron structures of GbBBX genes. The scale bar at the bottom of B, C, and D allows for the estimation of the length and relative position of each box.

**Figure 4 ijms-26-08427-f004:**
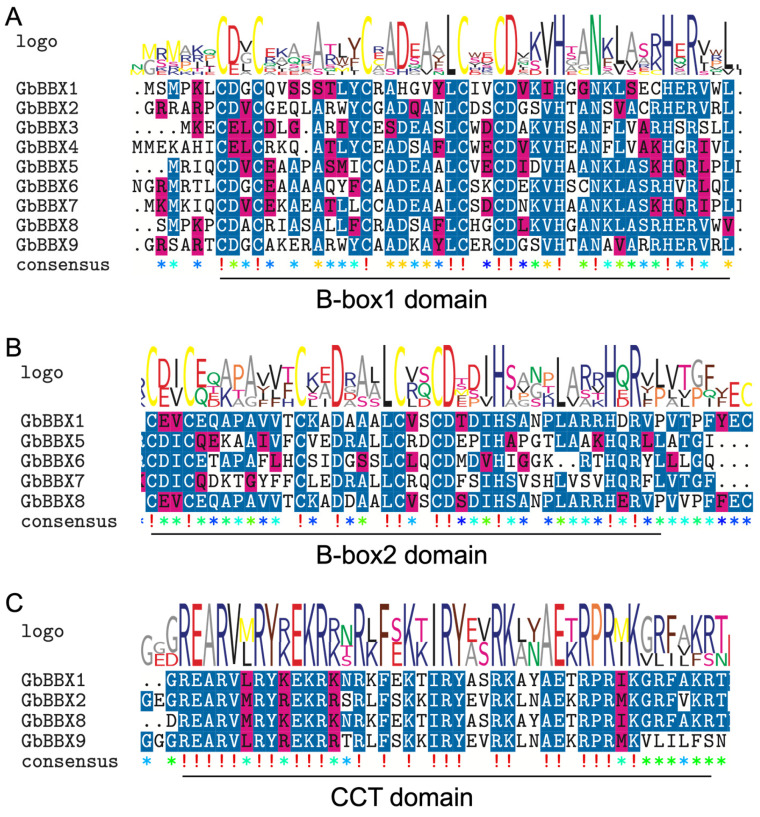
Seqlogo and sequence alignment of the B-box1, B-box2, and CCT domains of GbBBX proteins. (**A**) Multiple sequence alignment of the B-box1 domain. (**B**) Multiple sequence alignment of the B-box2 domain. (**C**) Multiple sequence alignment of the CCT domain. The seqlogo indicates the conservation rate of each amino acid; the sequence alignment reveals the conserved sequences within the domain. Identical conserved amino acids are highlighted in blue. Asterisks indicate non-conserved amino acid residues.

**Figure 5 ijms-26-08427-f005:**
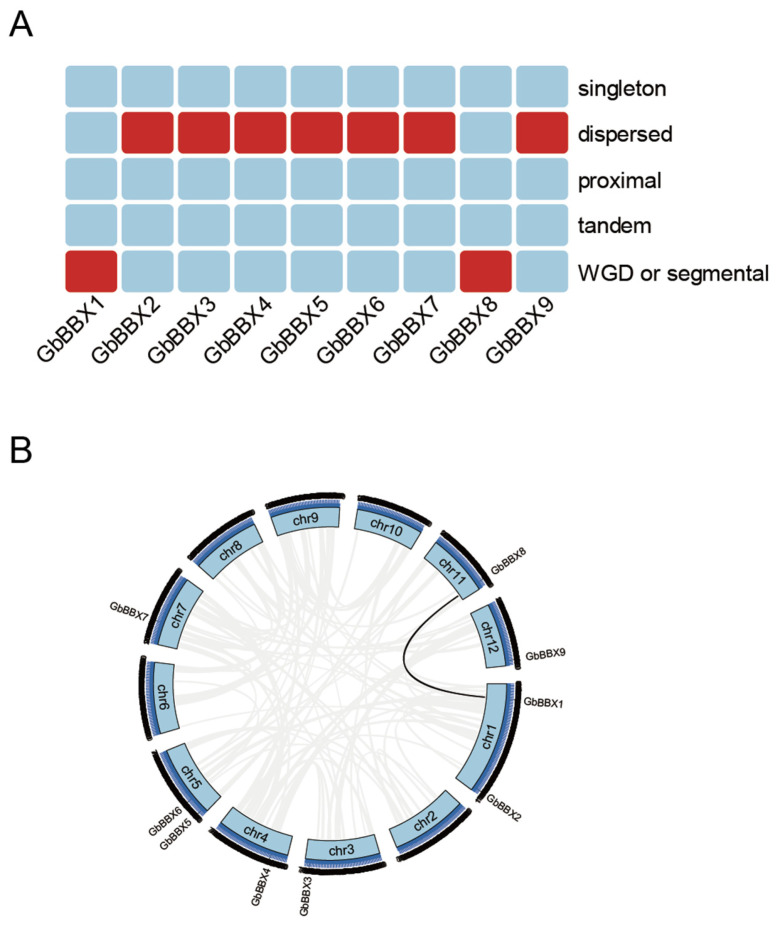
Evolutionary analysis of the *GbBBX* gene family. (**A**) The heatmap illustrates the duplication events (singleton, dispersed, proximal, tandem, and WGD or segmental duplication) of the *GbBBX* gene family. (**B**) Distribution and collinearity of the *GbBBX* gene family in the genome of *G. biloba*. The black line indicates the collinear gene pairs of *GbBBX* genes, while the grey line represents the collinear gene pairs in the *G. biloba* genome.

**Figure 6 ijms-26-08427-f006:**
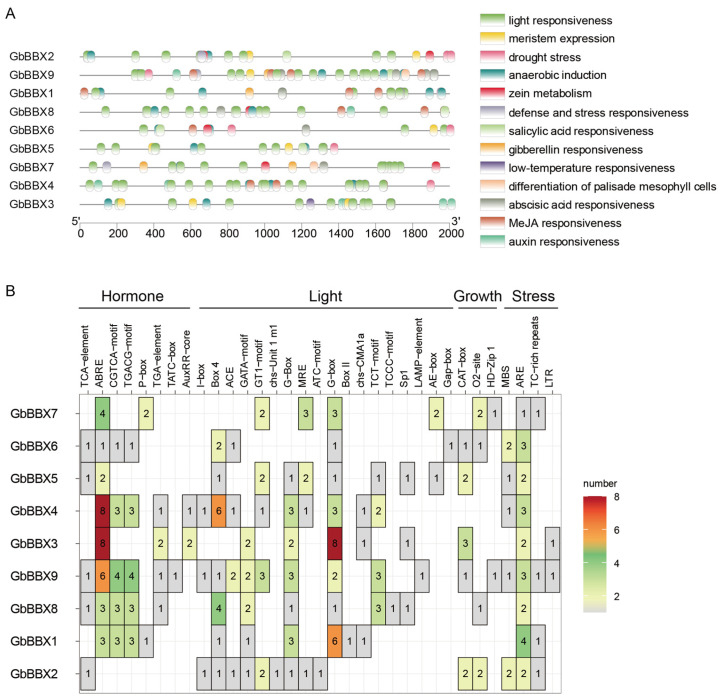
The cis-acting elements in the promoters of *GbBBX* genes. (**A**) The distribution of cis-acting elements in the promoters (upstream 2000 bp) of *GbBBX* genes. (**B**) The number of cis-acting elements among the four different types of response elements in the promoters of the *GbBBX* genes.

**Figure 7 ijms-26-08427-f007:**
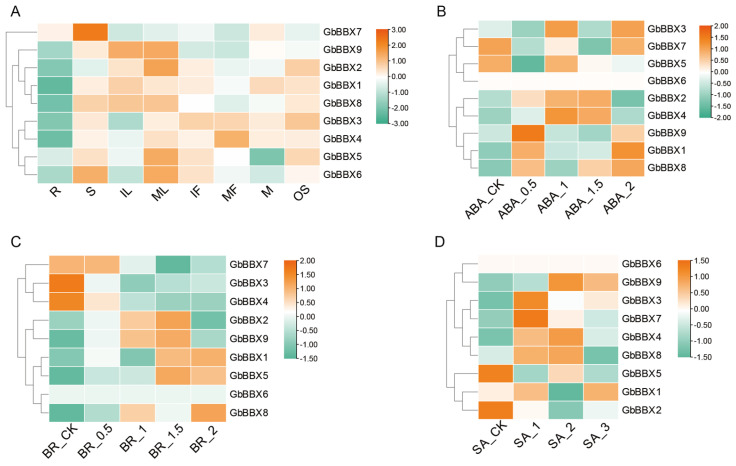
The heatmap illustrates the RNA-Seq-based expression patterns of 9 *GbBBX* genes across eight organs and under exogenous ABA, BR, and SA treatments. (**A**) Expression patterns of 9 *GbBBX* genes across eight organs, including root (R), stem (S), immature leaf (IL), mature leaf (ML), immature fruit (IF), mature fruit (MF), microstrobilus (M), and ovulate strobilus (OS). (**B**) Expression patterns of 9 *GbBBX* genes under exogenous ABA treatments. ABA_CK, 0 mmol/L concentration as the control (CK); ABA_0.5, 0.5 mmol/L concentration; ABA_1, 1 mmol/L concentration; ABA_1.5, 1.5 mmol/L concentration; ABA_2, 2 mmol/L concentration. (**C**) Expression patterns of 9 *GbBBX* genes under exogenous BR treatments. BR_CK, 0 mmol/L concentration as the control (CK); BR_0.5, 0.5 mmol/L concentration; BR_1, 1 mmol/L concentration; BR_1.5, 1.5 mmol/L concentration; BR_2, 2 mmol/L concentration. (**D**) Expression patterns of 9 *GbBBX* genes under exogenous SA treatments. SA_CK, 0 mmol/L concentration as the control (CK); SA_1, 1 mmol/L concentration; SA_2, 2 mmol/L concentration; SA_3, 3 mmol/L concentration. Red indicates high-expression genes, while blue represents low-expression genes. The heatmap was generated by the averages of three replicates.

**Figure 8 ijms-26-08427-f008:**
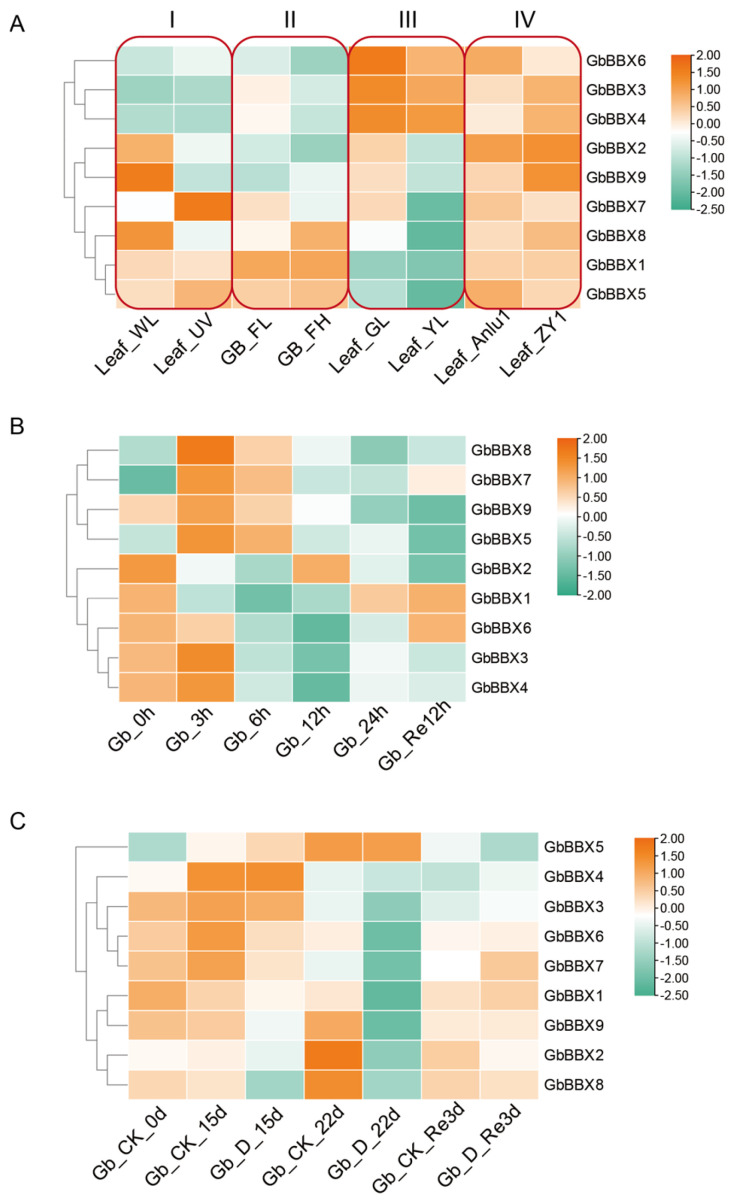
The heatmap illustrates the RNA-Seq-based expression patterns of 9 *GbBBX* genes in flavonoid biosynthesis and water stress. (**A**) Expression patterns of the 9 *GbBBX* genes grouped by contrasting low and high flavonoid content. Group I: Leaf_WL (UV-B-free white light, low flavonoid) vs. Leaf_UV (prolonged UV-B, high flavonoid); Group II: GB_FL (low-flavonoid clones) vs. GB_FH (high-flavonoid clones); Group III: Leaf_GL (green wild-type, low flavonoid) vs. Leaf_YL (yellow mutant, high flavonoid); Group IV: Leaf_Anlu1 (wild-type control) vs. Leaf_ZY1 (high-flavonoid mutant). (**B**) Expression patterns of 9 *GbBBX* genes under short-term water shock. 0 h, 3 h, 6 h, 12 h, and 24 h of dehydration and 12 h of rehydration. (**C**) Expression patterns of 9 *GbBBX* genes under long-term drought stress. Drought (D) for 0 d, 15 d, 22 d, and re-watering 3 d, compared with the control group (CK) with regular irrigation. Red indicates high-expression genes, while blue represents low-expression genes. The heatmap was generated by the averages of three replicates.

**Figure 9 ijms-26-08427-f009:**
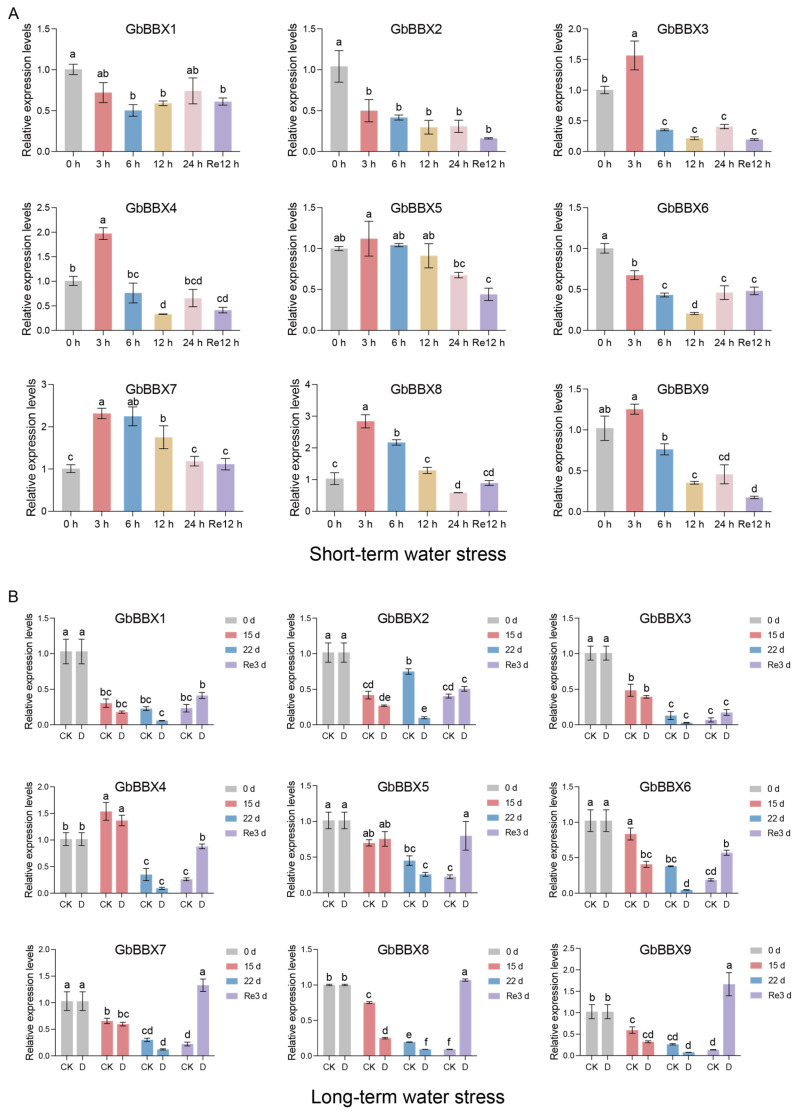
qRT-PCR analysis of *GbBBX* genes. (**A**) qRT-PCR analysis of *GbBBX* genes under short-term water shock. 0 h, 3 h, 6 h, 12 h, and 24 h of dehydration and 12 h of rehydration. (**B**) qRT-PCR analysis of *GbBBX* genes under long-term water stress. Drought (D) for 0 d, 15 d, 22 d, and re-watering 3 d, compared with the control group (CK) with regular irrigation. All data are presented as means ± SD (n = 3). Lowercase letters above bars denote statistically significant differences (*p* < 0.05) determined using one-way ANOVA with Tukey’s multiple comparisons test.

**Figure 10 ijms-26-08427-f010:**
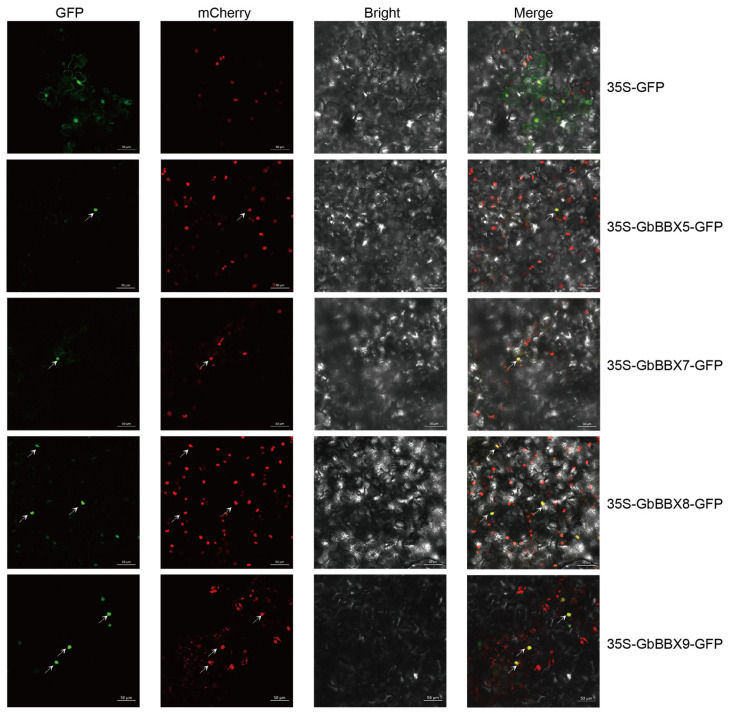
Subcellular localization of GbBBX genes. The GbBBX5-GFP, GbBBX7-GFP, GbBBX8-GFP, and GbBBX9-GFP fusions were transiently expressed in transgenic *Nicotiana benthamiana* leaves containing a nuclear localization protein with mCherry signals. The GFP and mCherry signals were visualized using confocal microscopy at 72 h after infiltration. Arrows highlight fluorescent signals in the micrograph.

**Table 1 ijms-26-08427-t001:** Basic information of *GbBBX* gene family members in *G. biloba*.

Gene Name	Gene ID	Peptide (aa)	Molecular Weight MW (Da)	Theoretical pI	Grand Average of Hydropathicity (GRAVY)	Instability Index	Aliphatic Index	Prediction of Subcellular Localization
GbBBX1	evm.model.chr1.520	434	48,292.50	5.19	−0.690	41.33	62.67	nuclear
GbBBX2	evm.model.chr1.3215	492	54,652.64	5.08	−0.684	44.93	63.82	nuclear
GbBBX3	evm.model.chr3.2337	248	26,808.51	4.84	−0.598	53.98	67.62	nuclear
GbBBX4	evm.model.chr4.597	376	42,009.76	8.11	−0.173	46.94	87.98	chloroplast
GbBBX5 (GbBBX25)	evm.model.chr5.165	273	29,581.52	4.96	−0.319	56.35	74.07	nuclear
GbBBX6	evm.model.chr5.546	265	29,541.79	7.83	−0.271	45.33	79.13	chloroplast
GbBBX7	evm.model.chr7.750	290	31,887.14	5.65	−0.434	51.32	75.38	nuclear
GbBBX8	evm.model.chr11.1562	412	45,117.02	5.80	−0.286	46.13	72.18	nuclear
GbBBX9	evm.model.chr12.1654	536	60,029.39	4.95	−0.746	47.55	66.08	nuclear

## Data Availability

The public transcriptome data used in this research can be accessed in the insert article.

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
