# Peer review of "Genome-Wide Identification and Expression Analysis of the *Ginkgo biloba* B-Box Gene Family in Response to Hormone Treatments, Flavonoid Levels, and Water Stress"

_ijms, 2025, doi:10.3390/ijms26178427_

Round 1

Reviewer 1 Report

Comments and Suggestions for Authors

This manuscript provides a comprehensive analysis of the BBX gene family in Ginkgo biloba, featuring detailed bioinformatics and quantitative analyses. The study is well-structured and offers valuable insights into the evolutionary and functional characteristics of these genes. However, some revisions are required before this manuscript can be accepted for publication.

The title's mention of "response to… flavonoid regulation" is methodologically inaccurate because the study employed materials with varying flavonoid contents rather than experimental flavonoid treatments.

Abstract

Line 26-27, “GbBBX genes contain motifs associated with light, hormones, and stress, suggesting their responses to flavonoid regulation”. This deduction seems rather strained, with the authors attempting to forcibly associate the results with flavonoids.

Is the BBX gene family limited to just 9 members in Ginkgo biloba. This appears to be an unusually low number compared to other species.

Results

9 BBX genes were identified (line126-127). And the authors show that “Following this, a gene with incomplete SBP domains was excluded based on information from the NCBI Conserved Domain Database (CDD).” (line 127-128). Consequently, there should be only eight BBX genes remaining.

After first mention, subsequent botanical Latin names should be abbreviated (e.g., Ginkgo biloba → G. biloba). Please carefully proofread and edit the entire manuscript.

Line 164-165, Why emphasize monocotyledon, dicotyledon, and gymnosperm in phylogenetic tree analysis.

The resolution of Figure 2 is too low to be clearly visible.

In subcellular localization experiments, why were only BBX5/7/8/9 tested, while the other five genes were not.

Discussion

The Discussion section should strengthen the functional analysis and speculation of BBX genes, as the current manuscript overly focuses on presenting results. Specifically, based on the BBX expression patterns, functional analysis and hypotheses should be further developed by integrating findings from studies in other species. Section “3.1. Characteristics and Evolution of BBX Genes in Ginkgo biloba” could be made more concise, while Section “3.2. Expression Patterns and Potential Functions of the GbBBX Genes” requires expansion with additional content.

Materials and Methods

The manuscript lacks detailed descriptions of statistical methods and data analysis procedures.

Author Response

Comments: This manuscript provides a comprehensive analysis of the BBX gene family in Ginkgo biloba, featuring detailed bioinformatics and quantitative analyses. The study is well-structured and offers valuable insights into the evolutionary and functional characteristics of these genes. However, some revisions are required before this manuscript can be accepted for publication.

Response: We greatly appreciate the positive comments and valuable suggestions from the reviewer. We feel great thanks for your professional review work on our article. According to the comments, we have carefully checked and addressed them in the revised manuscript. The point-to-point responses are as follows.

Comments 1: The title's mention of "response to… flavonoid regulation" is methodologically inaccurate because the study employed materials with varying flavonoid contents rather than experimental flavonoid treatments.

Response 1: Thanks for pointing this out. We have revised the title as “Genome-wide Identification and Expression Analysis of the Ginkgo biloba B-Box Gene Family in Response to Hormone Treatments, Flavonoid Levels, and Water Stress” in the manuscript, please see lines 2-4.

Comments 2: Line 26-27, “GbBBX genes contain motifs associated with light, hormones, and stress, suggesting their responses to flavonoid regulation”. This deduction seems rather strained, with the authors attempting to forcibly associate the results with flavonoids.

Response 2: Thanks for pointing this out. We have revised this sentence to “An examination of cis-regulatory elements indicated that numerous GbBBX genes contain motifs associated with light, hormones, and stress, suggesting their potential roles in responding to these signals and environmental adaptation” in the manuscript, please see line 26.

Comments 3: Is the BBX gene family limited to just 9 members in Ginkgo biloba. This appears to be an unusually low number compared to other species.

Response 3: Thanks for pointing this out. The BBX gene family in Ginkgo biloba was systematically analyzed utilizing two distinct methodologies: the BLAST approach and the HMMsearch approach. The BLAST method yielded the identification of 28 candidate genes (Supplementary Table S1), while the HMMsearch method identified only 10 candidate genes (Supplementary Table S1). Among these, nine genes were found to overlap between the two methods and were designated as GbBBX genes in Ginkgo biloba (Table 1). The genes with incomplete SBP domains were excluded based on information from the NCBI Conserved Domain Database (CDD). Therefore, our approach to gene family identification is robust and reliable, employing the most recently released high-quality genome assembly [1]. The lower gene numbers in Ginkgo biloba can be attributed to unique genomic characteristics: despite its large size (~9.87 Gb), repetitive sequences comprise ~65% of the genome. Consequently, the total number of predicted functional genes is considerably lower (~27,000) compared to model plants such as poplar (Populus alba x Populus tremula var. glandulosa clone 84K; ~85,000 genes; genome size: 747.5 Mb) [2]. This phenomenon of reduced gene number relative to genome size has been consistently observed across other transcription factor families in Ginkgo [3,4].

  1. Liu, H.; Wang, X.; Wang, G.; Cui, P.; Wu, S.; Ai, C.; Hu, N.; Li, A.; He, B.; Shao, X.; et al. The nearly complete genome of Ginkgo biloba illuminates gymnosperm evolution. Nat Plants 2021, 7, 748-756, doi:10.1038/s41477-021-00933-x.
  2. Qiu, D.; Bai, S.; Ma, J.; Zhang, L.; Shao, F.; Zhang, K.; Yang, Y.; Sun, T.; Huang, J.; Zhou, Y.; et al. The genome of Populus alba x Populus tremula var. glandulosa clone 84K. DNA Res 2019, 26, 423-431, doi:10.1093/dnares/dsz020.
  3. Liu, S.; Meng, Z.; Zhang, H.; Chu, Y.; Qiu, Y.; Jin, B.; Wang, L. Identification and characterization of thirteen gene families involved in flavonoid biosynthesis in Ginkgo biloba. Industrial Crops and Products 2022, 188, doi:10.1016/j.indcrop.2022.115576.
  4. Li, W.; Xiao, N.; Wang, Y.; Liu, X.; Chen, Z.; Gu, X.; Chen, Y. Genome-Wide Identification, Evolutionary and Functional Analyses of WRKY Family Members in Ginkgo biloba. Genes (Basel) 2023, 14, doi:10.3390/genes14020343.

Comments 4: 9 BBX genes were identified (line126-127). And the authors show that “Following this, a gene with incomplete SBP domains was excluded based on information from the NCBI Conserved Domain Database (CDD).” (line 127-128). Consequently, there should be only eight BBX genes remaining.

Response 4: We sincerely appreciate the reviewer’s meticulous attention to detail regarding the identification of BBX genes. Importantly, all nine candidate BBX genes identified in the initial screen were subjected to conserved domain analysis using the NCBI CDD. This analysis confirmed that each of these nine genes possesses complete SBP domains. Consequently, no genes were excluded based on incomplete domains at this stage. We apologize for any confusion caused by the mistaken wording and have revised this sentence in the revised manuscript, please see line 127.

Comments 5: After first mention, subsequent botanical Latin names should be abbreviated (e.g., Ginkgo biloba → G. biloba). Please carefully proofread and edit the entire manuscript.

Response 5: Thanks for pointing this out. We have proofread and revised “Ginkgo biloba” to “G. biloba” in the entire manuscript.

Comments 6: Line 164-165, Why emphasize monocotyledon, dicotyledon, and gymnosperm in phylogenetic tree analysis.

Response 6: Thanks for pointing this out. A dicotyledonous model plant (Arabidopsis thaliana), a monocotyledonous model plant (rice), and a gymnosperm (Ginkgo biloba), were selected for phylogenetic tree analysis. The phylogenetic tree results obtained from these plants are representative. We prioritized functional representativeness over numerical comprehensiveness because these species are established models with extensively annotated, high-quality genomes, and they cover fundamental evolutionary splits (gymnosperm/angiosperm, monocot/dicot). In addition, our study focuses on functional conservation/divergence in model systems rather than comprehensive gene family reconstruction.

Comments 7: The resolution of Figure 2 is too low to be clearly visible.

Response 7: We sincerely appreciate the reviewer's observation regarding the resolution of Figure 2 in the manuscript file. This issue occurs due to the automatic image compression applied when saving the document in Microsoft Word format. To ensure optimal clarity, we have provided a high-resolution vector PDF version of Figure 2 in the attachment (filename: Figures.zip). This scalable format retains full detail at any zoom level and resolves the visibility concerns. Should the editorial team require alternative file formats or additional adjustments, we are happy to provide them promptly.

Comments 8: In subcellular localization experiments, why were only BBX5/7/8/9 tested, while the other five genes were not.

Response 8: Thanks for pointing this out. This prioritization was driven by our study’s primary interest in water stress-responsive mechanisms. Expression profiling revealed that these four genes exhibited distinct differential expression patterns under short-term and long-term water stress conditions compared to other family members, suggesting their potential functional significance in this biological process. To efficiently focus experimental resources on candidates most relevant to our core research question, we selected these transcription factors for initial validation. Future studies will expand subcellular localization analysis to additional BBX genes as research progresses.

Comments 9: The Discussion section should strengthen the functional analysis and speculation of BBX genes, as the current manuscript overly focuses on presenting results. Specifically, based on the BBX expression patterns, functional analysis and hypotheses should be further developed by integrating findings from studies in other species. Section “3.1. Characteristics and Evolution of BBX Genes in Ginkgo biloba” could be made more concise, while Section “3.2. Expression Patterns and Potential Functions of the GbBBX Genes” requires expansion with additional content.

Response 9: Thank you for your nice suggestion. We have expanded Section 3.2 with additional content in the revised manuscript, please see the revised Section 3.2.

Comments 10: The manuscript lacks detailed descriptions of statistical methods and data analysis procedures.

Response 10: Thanks for pointing this out. We have added more details on statistical methods and data analysis procedures in the revised manuscript, please see lines 636-639.

Reviewer 2 Report

Comments and Suggestions for Authors

This study conducted a comprehensive genome-wide analysis of the BBX transcription factor family in Ginkgo biloba, identifying nine GbBBX genes and characterizing their roles in flavonoid biosynthesis and abiotic stress responses. It provides valuable insights into the evolution and functional diversification of BBX genes. The following revisions are required:

(1) Lines 17-18: "a lack of + investigations" contains a grammatical error. Please revise.
(2) Enhance the clarity of labels/text in Figure 3.
(3) Subcellular localization was validated for only four GbBBXs (5/7/8/9). Is localization data available for the other five GbBBXs?
(4) Figure 7: What was the duration of treatment for the different concentrations of ABA, SA, and BR?
(5) Figure 9A: Why are the Drought (D) and CK (control) treatments missing? We recommend adding significance indicators to Figure 9.
(6) Figure 9: Besides time-course dehydration, why was there no RT-qPCR validation for BBX expression under ABA, SA, and BR treatments?
(7) Line 131: Why does GbBBX5 have dual nomenclature?

(8)Line 140-143, modify language expression

Comments on the Quality of English Language

The manuscript requires professional English editing to enhance linguistic quality

Author Response

Comments: This study conducted a comprehensive genome-wide analysis of the BBX transcription factor family in Ginkgo biloba, identifying nine GbBBX genes and characterizing their roles in flavonoid biosynthesis and abiotic stress responses. It provides valuable insights into the evolution and functional diversification of BBX genes. The following revisions are required:

Response: We greatly appreciate the positive comments and valuable suggestions from the reviewer. We feel great thanks for your professional review work on our article. According to the comments, we have carefully checked and addressed them in the revised manuscript. The point-to-point responses are as follows.

Comments 1: Lines 17-18: "a lack of + investigations" contains a grammatical error. Please revise.

Response 1: Thanks for pointing this out. We have revised this sentence to “there has been a lack of systematic investigation into the BBX gene family in Ginkgo biloba” in the manuscript, please see line 17.

Comments 2: Enhance the clarity of labels/text in Figure 3.

Response 2: Thanks for pointing this out. This issue occurs due to the automatic image compression applied when saving the document in Microsoft Word format. To ensure optimal clarity, we have provided a high-resolution vector PDF version of Figure 3 in the attachment (filename: Figures.zip). This scalable format retains full detail at any zoom level and resolves the visibility concerns. Should the editorial team require alternative file formats or additional adjustments, we are happy to provide them promptly.

Comments 3: Subcellular localization was validated for only four GbBBXs (5/7/8/9). Is localization data available for the other five GbBBXs?

Response 3: Thanks for pointing this out. We are sorry that subcellular localization analysis was conducted exclusively for four GbBBX genes (GbBBX5, GbBBX7, GbBBX8, and GbBBX9), whereas the remaining five genes were not experimentally characterized in this investigation. This prioritization was driven by our study’s primary interest in water stress-responsive mechanisms. Expression profiling revealed that these four genes exhibited distinct differential expression patterns under short-term or long-term water stress conditions compared to other family members, suggesting their potential functional significance in this biological process. To efficiently focus experimental resources on candidates most relevant to our core research question, we selected these transcription factors for initial validation. Future studies will expand subcellular localization analysis to additional BBX genes as research progresses.

Comments 4: Figure 7: What was the duration of treatment for the different concentrations of ABA, SA, and BR?

Response 4: Thanks for pointing this out. Exogenous ABA and BR were sprayed on the leaf surface of one-year-old potted Ginkgo seedlings every 5 days, and mature leaves were harvested on day 20 after completing the five sprays [1]. SA was sprayed on the leaf surface of two-year-old potted Ginkgo seedlings. The spraying trials were conducted four times: June 8, 22, and 29 June 2019. Twenty days after completion of the 4 spray times, the mature leaves were collected for transcriptome sequencing [2].

  1. Guo, F.; Guo, J.; El-Kassaby, Y.A.; Wang, G. Genome-Wide Identification of Expansin Gene Family and Their Response under Hormone Exposure in Ginkgo biloba L. International Journal of Molecular Sciences 2023, 24, doi:10.3390/ijms24065901.
  2. Guo, F.; Xiong, W.; Guo, J.; Wang, G. Systematic Identification and Expression Analysis of the Auxin Response Factor (ARF) Gene Family in Ginkgo biloba L. International Journal of Molecular Sciences 2022, 23, doi:10.3390/ijms23126754.

Comments 5: Figure 9A: Why are the Drought (D) and CK (control) treatments missing? We recommend adding significance indicators to Figure 9.

Response 5: Thanks for pointing this out. The experimental design in Figure 9A represents a pseudotime-series analysis of progressive dehydration stress, not discrete drought/control treatments. The 0 h time point serves as the universal control (CK) for all subsequent dehydration intervals (3h, 6h, 12h, 24h) and 12 h of rehydration (Re12 h). In addition, we appreciate the reviewer's valuable recommendation regarding statistical annotation in Figure 9. This is indeed an excellent suggestion to enhance data interpretation. We have now incorporated significance indicators directly into Figure 9 panels based on one-way ANOVA with Tukey's post hoc multiple comparisons test (p < 0.05).

Comments 6: Figure 9: Besides time-course dehydration, why was there no RT-qPCR validation for BBX expression under ABA, SA, and BR treatments?

Response 6: Thanks for pointing this out. The transcriptome data for phytohormone treatments (ABA/SA/BR) were sourced exclusively from public databases [1, 2] where original biological samples were unavailable. Consequently, we were unable to perform experimental validation via RNA extraction and RT-qPCR for these specific conditions. In contrast, the dehydration time-course data and long-term water stress derive from our previously published study [3,4], wherein archived RNA samples remained accessible for targeted validation. This approach allowed us to prioritize experimental verification where original materials were obtainable while leveraging existing transcriptomic resources for hormone responses. We agree that direct validation of all treatments would strengthen the work and will address this in future functional studies.

  1. Guo, F.; Guo, J.; El-Kassaby, Y.A.; Wang, G. Genome-Wide Identification of Expansin Gene Family and Their Response under Hormone Exposure in Ginkgo biloba L. International Journal of Molecular Sciences 2023, 24, doi:10.3390/ijms24065901.
  2. Guo, F.; Xiong, W.; Guo, J.; Wang, G. Systematic Identification and Expression Analysis of the Auxin Response Factor (ARF) Gene Family in Ginkgo biloba L. International Journal of Molecular Sciences 2022, 23, doi:10.3390/ijms23126754.
  3. Meiling Ming; Juan Zhang; Jiamin Zhang; Jing Tang; Fangfang Fu; Cao., F. Transcriptome Profiling Identifies Plant Hormone Signaling Pathway-Related Genes and Transcription Factors in the Drought and Re-Watering Response of Ginkgo biloba. Plants 2024, 13, 2685, doi:10.3390/plants13192685.
  4. Meiling Ming; Juan Zhang; Jing Tang; Jiamin Zhang; Fangfang Fu; Cao, F. Transcriptome Profiling Revealed ABA Signaling Pathway-Related Genes and Major Transcription Factors Involved in the Response to Water Shock and Rehydration in Ginkgo biloba. forests 2024, 15(10), 1690, doi:10.3390/f15101690.

Comments 7: Line 131: Why does GbBBX5 have dual nomenclature?

Response 7: Thanks for pointing this out. In the present study, we identified nine BBX genes within the G. biloba reference genome and renamed them as GbBBX1-9 according to their chromosomal locations (Table 1). However, there was a GbBBX5 (also designated as GbBBX25) that has been demonstrated to be involved in salt tolerance in the previous study. It was named GbBBX25 due to its high homology with Arabidopsis thaliana AtBBX25 in that study [1]. Dual nomenclature was assigned to this particular BBX gene (GbBBX5/GbBBX25) as it represents the sole functionally characterized member within the Ginkgo biloba BBX family to date. This designation facilitates reader comprehension by maintaining consistency with previously published functional studies while aligning with our systematic gene numbering convention.

  1. Huang, S.; Chen, C.; Xu, M.; Wang, G.; Xu, L.A.; Wu, Y. Overexpression of Ginkgo BBX25 enhances salt tolerance in Transgenic Populus. Plant Physiol Biochem 2021, 167, 946-954, doi:10.1016/j.plaphy.2021.09.021.

Comments 8: Line 140-143, modify language expression

Response 8: Thanks for pointing this out. We have modified the language expression of this sentence in the revised manuscript, please see lines 140-141.

Reviewer 3 Report

Comments and Suggestions for Authors

The abstract states “there has been a lack of systematic investigations of the BBX gene family in Ginkgo biloba,” but similar studies in gymnosperms or related woody species could be briefly acknowledged to strengthen novelty positioning.

Lines 134–137-The instability indices of all GbBBX proteins are reported to be above 40 (thus unstable). Please discuss the potential biological implications or limitations of interpreting this, given that many plant proteins with regulatory functions often fall into “unstable” classifications in silico.

Lines 147–149-The chromosomal distribution description could benefit from a figure reference (i.e., “as shown in Figure 1”) to improve readability.

Lines 172–188-In motif analysis, please explain more clearly how the functional relevance of motif differences across groups was inferred. Are there known functional studies supporting such associations?

Lines 221–225-In the multiple sequence alignment section, while consensus sequences are nicely provided, consider quantifying sequence identity percentages for the B-box and CCT domains to strengthen conservation claims.

Lines 257–279-The cis-element analysis is quite comprehensive. Could you indicate which specific GbBBX genes had the highest number of hormone or light response elements? This might direct future functional studies.

Lines 332–351-When describing flavonoid-related expression profiles, it would be useful to include a clearer interpretation of why there is limited overlap of differentially expressed GbBBX genes among the four groups. Does this suggest condition-specific regulation or redundancy?

Lines 388–394-The qRT-PCR validation for long-term drought shows that GbBBX5 increases at 15 days then decreases by 22 days. This dynamic is interesting—could you discuss possible physiological interpretations (e.g., early vs. late stress signaling roles)?

The discussion would benefit from a short paragraph comparing the number of BBX genes in Ginkgo versus other gymnosperms (if data exist), not just Arabidopsis or rice, to better frame evolutionary perspectives.

Throughout the methods section, please include software versions and precise parameters used for MEME (number of motifs, e-value threshold) and PlantCARE (default or adjusted settings), for reproducibility.

In the subcellular localization section, you mention co-localization with NLS-mCherry. Quantitative co-localization statistics (like Pearson’s correlation coefficient) could strengthen this point.

It would be helpful to briefly mention in the conclusion any foreseeable applications of these results in Ginkgo breeding programs (e.g., improving drought resistance or flavonoid content).

The manuscript is generally well written, but the long complex sentences, especially in the introduction and discussion, could be split for clarity. Consider additional short paragraphs for major sub-themes.

Comments on the Quality of English Language

The manuscript is generally clear and well-structured, there are some instances of overly long sentences and somewhat cumbersome phrasing, particularly in the Introduction and Discussion sections. Shortening sentences and improving transitions would enhance readability. Additionally, minor grammatical issues occur throughout the text, which should be corrected. 

Author Response

Comments 1: The abstract states “there has been a lack of systematic investigations of the BBX gene family in Ginkgo biloba,” but similar studies in gymnosperms or related woody species could be briefly acknowledged to strengthen novelty positioning.

Response 1: We sincerely appreciate the reviewer's insightful suggestion regarding novelty positioning. As recommended, we have now acknowledged prior studies of BBX genes in related woody species in the revised Introduction (Lines 103-104). These additions clarify that while BBX genes have been characterized in some woody species, a systematic investigation specifically targeting Ginkgo biloba—a gymnosperm with a unique evolutionary status—remains unreported.

Comments 2: Lines 134–137-The instability indices of all GbBBX proteins are reported to be above 40 (thus unstable). Please discuss the potential biological implications or limitations of interpreting this, given that many plant proteins with regulatory functions often fall into “unstable” classifications in silico.

Response 2: We acknowledge the reviewer's valuable point regarding the predicted instability indices (>40) of all GbBBX proteins. While in silico instability predictions align with the paradigm that regulatory proteins like transcription factors often exhibit short half-lives—enabling rapid turnover for dynamic stress response modulation—we emphasize that such computational models have inherent limitations in capturing in vivo stabilization mechanisms. These include protein-protein interactions (e.g., dimerization), post-translational modifications (phosphorylation/SUMOylation), chaperone associations, and subcellular compartmentalization, all of which can confer functional stability despite high instability scores.

Comments 3: Lines 147–149-The chromosomal distribution description could benefit from a figure reference (i.e., “as shown in Figure 1”) to improve readability.

Response 3: Thanks for pointing this out. We have added the figure reference to improve readability in the revised manuscript, please see lines 145-147.

Comments 4: Lines 172–188-In motif analysis, please explain more clearly how the functional relevance of motif differences across groups was inferred. Are there known functional studies supporting such associations?

Response 4: Thanks for pointing this out. The differential functions across subgroups are primarily mediated by conserved domains—directly determined by motif compositions—which dictate distinct molecular mechanisms.

Comments 5: Lines 221–225-In the multiple sequence alignment section, while consensus sequences are nicely provided, consider quantifying sequence identity percentages for the B-box and CCT domains to strengthen conservation claims.

Response 5: Thanks for pointing this out. We confirm that in the sequence logos (Figure 4), the height of each letter directly represents positional conservation strength, with taller letters indicating higher conservation at that residue position.

Comments 6: Lines 257–279-The cis-element analysis is quite comprehensive. Could you indicate which specific GbBBX genes had the highest number of hormone or light response elements? This might direct future functional studies.

Response 6: Thanks for your nice suggestion. GbBBX9 contained the highest density of cis-regulatory elements associated with hormone, light, and stress responsiveness, whereas GbBBX2 featured the most abundant elements linked to growth regulation. We have added more details in the revised manuscript, please see lines 277-279.

Comments 7: Lines 332–351-When describing flavonoid-related expression profiles, it would be useful to include a clearer interpretation of why there is limited overlap of differentially expressed GbBBX genes among the four groups. Does this suggest condition-specific regulation or redundancy?

Response 7: Thanks for your nice suggestion. We have added more discussion in the revised manuscript, please see lines 515-525. The distinct sets of differentially expressed GbBBX genes observed across the four experimental groups might stem from fundamental differences in their underlying biological perturbations, which activate discrete regulatory pathways: Group I (Light Quality Effect): Leaf_WL vs Leaf_UV isolates UV-B photoreceptor signaling (GbBBX2, GbBBX8, and GbBBX9). Group II (Genetic Variation Effect): GB_FL vs GB_FH across 112 ginkgo clones captures constitutive genetic polymorphisms in flavonoid regulators. Group III (Developmental Mutation Effect): Leaf_GL vs Leaf_YL compares wild-type and chlorophyll-deficient mutants, where DEGs (GbBBX2, GbBBX8, and GbBBX9) correlate with retrograde signaling from plastid-to-nucleus—a mechanism absent in other groups. Group IV (Artificial Selection Effect): Leaf_ZY1 (high-flavonoid mutant) vs Leaf_Anlu1 highlights breeding-selected transcriptional rewiring (GbBBX3 and GbBBX4). This compartmentalization confirms that GbBBX genes operate within stimulus-specific regulatory modules: photoreception (Group I), genetic background (Group II), organelle-nucleus crosstalk (Group III), and artificial selection (Group IV).

Comments 8: Lines 388–394-The qRT-PCR validation for long-term drought shows that GbBBX5 increases at 15 days then decreases by 22 days. This dynamic is interesting—could you discuss possible physiological interpretations (e.g., early vs. late stress signaling roles)?

Response 8: We sincerely appreciate the reviewer's insightful observation regarding the GbBBX5 expression dynamics under long-term drought. Upon re-examining the qRT-PCR data with rigorous statistical validation (Tukey's multiple comparisons test, *p* < 0.05), the apparent temporal pattern (increase at 15d then decrease at 22d) was found to lack statistical significance due to overlapping error bars between CK and drought. Consequently, we have removed this specific temporal claim in the revised manuscript. We apologize for the oversight in initial data interpretation and thank the reviewer for prompting this important correction.

Comments 9: The discussion would benefit from a short paragraph comparing the number of BBX genes in Ginkgo versus other gymnosperms (if data exist), not just Arabidopsis or rice, to better frame evolutionary perspectives.

Response 9: We sincerely appreciate the reviewer’s valuable suggestion regarding comparative analysis of BBX gene numbers across gymnosperms. To our knowledge, no genome-wide identification of BBX genes has been reported in non-ginkgophyte gymnosperms, creating a methodological gap for direct cross-species comparisons.

Comments 10: Throughout the methods section, please include software versions and precise parameters used for MEME (number of motifs, e-value threshold) and PlantCARE (default or adjusted settings), for reproducibility.

Response 10: Thanks for pointing this out. We have added software versions and precise parameters used for MEME and PlantCARE in the revised manuscript, please see the methods section.

Comments 11: In the subcellular localization section, you mention co-localization with NLS-mCherry. Quantitative co-localization statistics (like Pearson’s correlation coefficient) could strengthen this point.

Response 11: We appreciate the reviewer's suggestion regarding quantitative co-localization analysis. In standard subcellular localization protocols, visual confirmation of signal overlap between fluorescent tags (e.g., GFP-tagged targets and NLS-mCherry controls) is universally accepted as sufficient evidence for nuclear localization. Pearson's correlation coefficients are typically reserved for cases with partial co-localization (e.g., organelle-membrane interfaces) or when quantifying protein complex distributions—neither of which applies to our clear nuclear localization results. Nevertheless, should the reviewer consider quantitative metrics essential, we would be happy to incorporate Pearson's correlation analyses in the revised manuscript. High-resolution confocal stacks are archived for immediate reanalysis.

Comments 12: It would be helpful to briefly mention in the conclusion any foreseeable applications of these results in Ginkgo breeding programs (e.g., improving drought resistance or flavonoid content).

Response 12: Thanks for your nice suggestion. We have briefly mentioned in the revised conclusion any foreseeable applications of these results in Ginkgo breeding programs, please see lines 653-655.

Comments 13: The manuscript is generally well written, but the long complex sentences, especially in the introduction and discussion, could be split for clarity. Consider additional short paragraphs for major sub-themes.

Response 13: We sincerely appreciate the reviewer's valuable feedback regarding sentence structure and paragraph organization. As suggested, we have carefully reviewed the manuscript—particularly the Introduction and Discussion sections—and implemented revisions to enhance readability. This involved splitting lengthy sentences into more concise constructions, restructuring complex paragraphs around focused sub-themes, and ensuring logical flow while rigorously preserving the original scientific meaning and technical accuracy. These edits have been applied throughout the revised manuscript to improve clarity without compromising content integrity.

Reviewer 4 Report

Comments and Suggestions for Authors

The authors addressed on the Ginkgo biloba B-Box genes for the first time, which has impact on the community. However, some points should be addressed for the publication.

1. Several typos (gene names and species names should be in italic or statement on the result should be in past tense) are found in line 79, 219, 263, 292 and so on. Please fix those.

2. In line 131-132, authors mentioned on GbBBX5 has been demonstrated to be involved in salt tolerance while GbBBX families are supposed to be first analyzed in this study. What is the evidence of this statement? If the authors do not have the appropriate evidence, this sentense should be removed.

3. In figure 2, clade I and III seems to share the common branch, same was also shown in IV and V. Is this result agree with the Arabidopsis study in the past or is a noble finding or error in this study? Also, is not expected for the GbBBX genes to be located in the ancestral or at least in the lineage of potentially ancestral branch? However, GbBBX8 seems to be located alongside the evolutionary tract. What is the author's opinion on this?

4. Line 440, it was hard to understand that the distributions in two different organisms are parallels. Please rewrite them into scientific language with details.

5. Line 479-480, data could mean that but also could mean another scenarios. Authors are removing the alternative evolutionary scenarios without reason. Either to re-write them with indicating the feasible scenarios such as 'distinct evolutionary tract' or deleting the sentence will be adequate.

Author Response

Comments: The authors addressed on the Ginkgo biloba B-Box genes for the first time, which has impact on the community. However, some points should be addressed for the publication.

Response: We greatly appreciate the positive comments and valuable suggestions from the reviewer. We feel great thanks for your professional review work on our article. According to the comments, we have carefully checked and addressed them in the revised manuscript. The point-to-point responses are as follows.

Comments 1: Several typos (gene names and species names should be in italic or statement on the result should be in past tense) are found in line 79, 219, 263, 292 and so on. Please fix those.

Response 1: Thanks for pointing this out. We have revised these typos in the manuscript.

Comments 2: In line 131-132, authors mentioned on GbBBX5 has been demonstrated to be involved in salt tolerance while GbBBX families are supposed to be first analyzed in this study. What is the evidence of this statement? If the authors do not have the appropriate evidence, this sentense should be removed.

Response 2: Thanks for pointing this out. In the present study, we identified nine BBX genes within the G. biloba reference genome and renamed them as GbBBX1-9 according to their chromosomal locations (Table 1). However, there was a GbBBX25  that has been demonstrated to be involved in salt tolerance in the previous study. It was named GbBBX25 due to its high homology with Arabidopsis thaliana AtBBX25 in that study [1]. Dual nomenclature was assigned to this particular BBX gene (GbBBX5/GbBBX25) as it represents the sole functionally characterized member within the Ginkgo biloba BBX family to date. This designation facilitates reader comprehension by maintaining consistency with previously published functional studies while aligning with our systematic gene numbering convention.

  1. Huang, S.; Chen, C.; Xu, M.; Wang, G.; Xu, L.A.; Wu, Y. Overexpression of Ginkgo BBX25 enhances salt tolerance in Transgenic Populus. Plant Physiol Biochem 2021, 167, 946-954, doi:10.1016/j.plaphy.2021.09.021.

Comments 3: In figure 2, clade I and III seems to share the common branch, same was also shown in IV and V. Is this result agree with the Arabidopsis study in the past or is a noble finding or error in this study? Also, is not expected for the GbBBX genes to be located in the ancestral or at least in the lineage of potentially ancestral branch? However, GbBBX8 seems to be located alongside the evolutionary tract. What is the author's opinion on this?

Response 3: Thanks for pointing this out. The shared branch between Clade I/III and Clade IV/V is not an analytical error but reflects evolutionarily conserved domain architectures, consistent with established studies in Arabidopsis and other species. In Arabidopsis, Group I (dual B-box + CCT) and Group III (single B-box + CCT) share the CCT domain, which enables nuclear localization and DNA binding, while Group IV (dual B-box) and Group V (single B-box) lack CCT but retain protein interaction capabilities via B-box domains. This structural dichotomy explains the deeper branching split between the "CCT-containing" (I/III) and "CCT-lacking" (IV/V) superclades observed in our analysis, mirroring Arabidopsis phylogenies where Group I/III and IV/V form sister clades. The basal position of GbBBX8 may reflect Ginkgo's status as a paleoendemic species preserving ancestral BBX forms, with implications for reconstructing stress-response mechanisms in seed plants.

Comments 4: Line 440, it was hard to understand that the distributions in two different organisms are parallels. Please rewrite them into scientific language with details.

Response 4: Thanks for pointing this out. We have rewritten this sentence in the revised manuscript, please see line 439.

Comments 5: Line 479-480, data could mean that but also could mean another scenarios. Authors are removing the alternative evolutionary scenarios without reason. Either to re-write them with indicating the feasible scenarios such as 'distinct evolutionary tract' or deleting the sentence will be adequate.

Response 5: Thanks for your nice suggestion. We have deleted this sentence in the revised manuscript, please see line 479.